# Defining and Characterizing Reward Hacking

**Joar Skalse**[*]
University of Oxford

**Nikolaus H. R. Howe**
Mila, Université de Montréal

**Dmitrii Krasheninnikov**
University of Cambridge

**David Krueger**[*]
University of Cambridge

## Abstract

We provide the first formal definition of **reward hacking**, a phenomenon where optimizing an imperfect proxy reward function, $\tilde{\mathcal{R}}$, leads to poor performance according to the true reward function, $\mathcal{R}$. We say that a proxy is **unhackable** if increasing the expected proxy return can never decrease the expected true return. Intuitively, it might be possible to create an unhackable proxy by leaving some terms out of the reward function (making it "narrower") or overlooking fine-grained distinctions between roughly equivalent outcomes, but we show this is usually not the case. A key insight is that the linearity of reward (in state-action visit counts) makes unhackability a very strong condition. In particular, for the set of all stochastic policies, two reward functions can only be unhackable if one of them is constant. We thus turn our attention to deterministic policies and finite sets of stochastic policies, where non-trivial unhackable pairs always exist, and establish necessary and sufficient conditions for the existence of simplifications, an important special case of unhackability. Our results reveal a tension between using reward functions to specify narrow tasks and aligning AI systems with human values.

## 1 Introduction

It is well known that optimising a proxy can lead to unintended outcomes: a boat spins in circles collecting "powerups" instead of following the race track in a racing game (Clark and Amodei, 2016); an evolved circuit listens in on radio signals from nearby computers' oscillators instead of building its own (Bird and Layzell, 2002); universities reject the most qualified applicants in order to appear more selective and boost their ratings (Golden, 2001). In the context of reinforcement learning (RL), such failures are called **reward hacking**.

For AI systems that take actions in safety-critical real world environments such as autonomous vehicles, algorithmic trading, or content recommendation systems, these unintended outcomes can be catastrophic. This makes it crucial to align autonomous AI systems with their users' intentions. Precisely specifying which behaviours are or are not desirable is challenging, however. One approach to this specification problem is to learn an approximation of the true reward function (Ng et al., 2000; Ziebart, 2010; Leike et al., 2018). Optimizing a learned proxy reward can be dangerous, however; for instance, it might overlook side-effects (Krakovna et al., 2018; Turner et al., 2019) or encourage power-seeking (Turner et al., 2021) behavior. This raises the question motivating our work: When is it safe to optimise a proxy?

To begin to answer this question, we consider a somewhat simpler one: When *could* optimising a proxy lead to worse behaviour? "Optimising", in this context, does not refer to finding a global, or even local, optimum, but rather running a search process, such as stochastic gradient descent

---

[*]Equal contribution. Correspondence to: `joar.mvs@gmail.com, david.scott.krueger@gmail.com`

36th Conference on Neural Information Processing Systems (NeurIPS 2022).

(SGD), that yields a sequence of candidate policies, and tends to move towards policies with higher (proxy) reward. We make no assumptions about the path through policy space that optimisation takes.[1] Instead, we ask whether there is *any* way in which improving a policy according to the proxy could make the policy worse according to the true reward; this is equivalent to asking if there exists a pair of policies $\pi_1, \pi_2$ where the proxy prefers $\pi_1$, but the true reward function prefers $\pi_2$. When this is the case, we refer to this pair of true reward function and proxy reward function as **hackable**.

Given the strictness of our definition, it is not immediately apparent that any non-trivial examples of unhackable reward function pairs exist. And indeed, if we consider the set of all stochastic policies, they do not (Section 5.1). However, restricting ourselves to *any* finite set of policies guarantees at least one non-trivial unhackable pair (Section 5.2).

Intuitively, we might expect the proxy to be a "simpler" version of the true reward function. Noting that the definition of unhackability is symmetric, we introduce the asymmetric special case of **simplification**, and arrive at similar theoretical results for this notion.[2] In the process, and through examples, we show that seemingly natural ways of simplifying reward functions often fail to produce simplifications in our formal sense, and in fact fail to rule out the potential for reward hacking.

We conclude with a discussion of the implications and limitations of our work. Briefly, our work suggests that a proxy reward function must satisfy demanding standards in order for it to be safe to optimize. This in turn implies that the reward functions learned by methods such as reward modeling and inverse RL are perhaps best viewed as auxiliaries to policy learning, rather than specifications that should be optimized. This conclusion is weakened, however, by the conservativeness of our chosen definitions; future work should explore when hackable proxies can be shown to be safe in a probabilistic or approximate sense, or when subject to only limited optimization.

## 2 Example: Cleaning Robot

Consider a household robot tasked with cleaning a house with three rooms: Attic ⬛A, Bedroom ⬛, and Kitchen ⬛. The robot's (deterministic) policy is a vector indicating which rooms it cleans: $\pi = [\pi_1, \pi_2, \pi_3] \in \{0, 1\}^3$. The robot receives a (non-negative) reward of $r_1, r_2, r_3$ for cleaning the attic, bedroom, and kitchen, respectively, and the total reward is given by $J(\pi) = \pi \cdot r$. For example, if $r = [1, 2, 3]$ and the robot cleans the attic and the kitchen, it receives a reward of $1 + 3 = 4$.

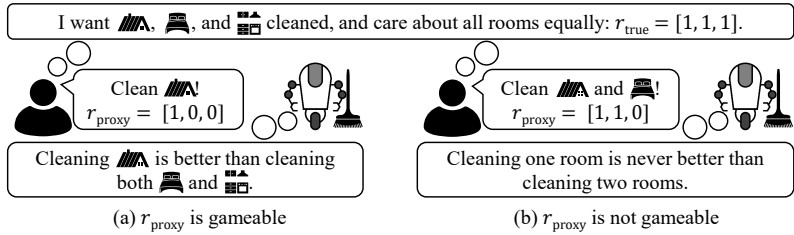

(a) $r_{\text{proxy}}$ is gameable  (b) $r_{\text{proxy}}$ is not gameable

Figure 1: An illustration of hackable and unhackable proxy rewards arising from overlooking rewarding features. A human wants their house cleaned. In (a), the robot draws an incorrect conclusion because of the proxy; this could lead to hacking. In (b), no such hacking can occur: the proxy is unhackable.

At least two ideas come to mind when thinking about "simplifying" a reward function. The first one is *overlooking rewarding features*: suppose the true reward is equal for all the rooms, $r_{\text{true}} = [1, 1, 1]$, but we only ask the robot to clean the attic and bedroom, $r_{\text{proxy}} = [1, 1, 0]$. In this case, $r_{\text{proxy}}$ and $r_{\text{true}}$ are unhackable. However, if we ask the robot to only clean the attic, $r_{\text{proxy}} = [1, 0, 0]$, this is hackable with respect to $r_{\text{true}}$. To see this, note that according to $r_{\text{proxy}}$ cleaning the attic ($J_{\text{proxy}} = 1$) is better than cleaning the bedroom and the kitchen ($J_{\text{proxy}} = 0$). Yet, $r_{\text{true}}$ says that cleaning the attic ($J_{\text{true}} = 1$) is worse than cleaning the bedroom and the kitchen ($J_{\text{true}} = 2$). This situation is illustrated in Figure 1.

---

[1]This assumption – although conservative – is reasonable because optimisation in state-of-the-art deep RL methods is poorly understood and results are often highly stochastic and suboptimal.

[2]See Section 4.2 for formal definitions.

The second seemingly natural way to simplify a reward function is *overlooking fine details*: suppose $r_{\text{true}} = [1, 1.5, 2]$, and we ask the robot to clean all the rooms, $r_{\text{proxy}} = [1, 1, 1]$. For these values, the proxy and true reward are unhackable. However, with a slightly less balanced true reward function such as $r_{\text{true}} = [1, 1.5, 3]$ the proxy does lead to hacking, since the robot would falsely calculate that it's better to clean the attic and the bedroom than the kitchen alone.

These two examples illustrate that while simplification of reward functions is sometimes possible, attempts at simplification can easily lead to reward hacking. Intuitively, omitting/overlooking details is okay so long as all these details are not as important together as any of the details that we do share. In general, it is not obvious what the proxy must look like to avoid reward hacking, suggesting we should take great care when using proxies. For this specific environment, a proxy and a true reward are hackable exactly when there are two sets of rooms $S_1, S_2$ such that the true reward gives strictly higher value to cleaning $S_1$ than it does to cleaning $S_2$, and the proxy says the opposite: $J_1(S_1) > J_1(S_2)$ & $J_2(S_1) < J_2(S_2)$. For a proof of this statement, see Appendix D.2.1.

## 3 Related Work

While we are the first to define hackability, we are far from the first to study specification hacking. The observation that optimizing proxy metrics tends to lead to perverse instantiations is often called "Goodhart's Law", and is attributed to Goodhart (1975). Manheim and Garrabrant (2018) provide a list of four mechanisms underlying this observation.

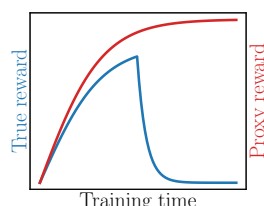

Figure 2: An illustration of reward hacking when optimizing a hackable proxy. The true reward first increases and then drops off, while the proxy reward continues to increase.

Examples of such unintended behavior abound in both RL and other areas of AI; Krakovna et al. (2020) provide an extensive list. Notable recent instances include a robot positioning itself between the camera and the object it is supposed to grasp in a way that tricks the reward model (Amodei et al., 2017), the previously mentioned boat race example (Clark and Amodei, 2016), and a multitude of examples of reward model hacking in Atari (Ibarz et al., 2018). Reward hacking can occur suddenly. Ibarz et al. (2018) and Pan et al. (2022) showcase plots similar to one in Figure 2, where optimizing the proxy (either a learned reward model or a hand-specified reward function) first leads to both proxy and true rewards increasing, and then to a sudden phase transition where the true reward collapses while the proxy continues going up.

Note that not all of these examples correspond to optimal behavior according to the proxy. Indeed, convergence to suboptimal policies is a well-known issue in RL (Thrun and Schwartz, 1993). As a consequence, improving optimization often leads to unexpected, qualitative changes in behavior. For instance, Zhang et al. (2021) demonstrate a novel cartwheeling behavior in the widely studied Half-Cheetah environment that exceeds previous performance so greatly that it breaks the simulator. The unpredictability of RL optimization is a key motivation for our definition of hackability, since we cannot assume that agents will find an optimal policy. Neither can we rule out the possibility of sudden improvements in proxy reward and corresponding qualitative changes in behavior. Unhackability could provide confidence that reward hacking will not occur despite these challenges.

Despite the prevalence and potential severity of reward hacking, to our knowledge Pan et al. (2022) provide the first peer-reviewed work that focuses specifically on it, although Everitt et al. (2017) tackle the closely related issue of reward corruption. The work of Pan et al. (2022) is purely empirical; they manually construct proxy rewards for several diverse environments, and evaluate whether optimizing these proxies leads to reward hacking; in 5 out of 9 of their settings, it does. In another closely related work, Zhuang and Hadfield-Menell (2020) examine what happens when the proxy reward function depends on a strict subset of features relevant for the true reward. They show that optimizing the proxy reward can lead to arbitrarily low true reward under suitable assumptions. This can be seen as a seemingly valid simplification of the true reward that turns out to be (highly) hackable. While their result only applies to environments with decreasing marginal utility and increasing opportunity cost, we demonstrate hackability is an issue in arbitrary MDPs.

Hackability is particularly concerning given arguments that reward optimizing behavior tends to be power-seeking (Turner et al., 2021). But Leike et al. (2018) establish that any desired behavior (power-seeking or not) can in principle be specified as optimal via a reward function.[3] However, unlike us, they do not consider the entire policy preference ordering. Meanwhile, Abel et al. (2021) note that Markov reward functions cannot specify arbitrary orderings over policies or trajectories, although they do not consider hackability. Previous works consider reward functions to be equivalent if they preserve the ordering over policies (Ng et al., 1999, 2000). Unhackability relaxes this, allowing equalities to be refined to inequalities, and vice versa. Unhackability provides a notion of what it means to be "aligned enough"; Brown et al. (2020b) provide an alternative. They say a policy is $\varepsilon$-value aligned if its value at every state is close enough to optimal (according to the true reward function). Neither notion implies the other.

*Reward tampering* (Everitt et al., 2017; Kumar et al., 2020; Uesato et al., 2020; Everitt et al., 2021) can be viewed as a special case of reward hacking, and refers to an agent corrupting the process generating reward signals, e.g. by tampering with sensors, memory registers storing the reward signal, or other hardware. Everitt et al. (2017) introduce the Corrupt Reward MDP (CRMDP), to model this possibility. A CRMDP distinguishes corrupted and uncorrupted rewards; these are exactly analogous to the proxy and true reward discussed in our work and others. Leike et al. (2018) distinguish reward tampering from *reward gaming*, where an agent achieves inappropriately high reward without tampering. However, in principle, a reward function could prohibit all forms of tampering if the effects of tampering are captured in the state. So this distinction is somewhat imprecise, and the CRMDP framework is general enough to cover both forms of hacking.

Our notion of simplification bears a close resemblance to quantilization (Taylor, 2016). Quantilization returns a random policy from the top n% best policies. This is similar to equating the values of those policies, but a simplification may also equate the values of the bottom/middle n%, etc. Thus simplification may achieve a similar effect to quantilization without assuming that we are free to choose from among the best policies.

## 4    Preliminaries

We begin with an overview of reinforcement learning (RL) to establish our notation and terminology. Section 4.2 introduces our novel definitions of hackability and simplification.

### 4.1    Reinforcement Learning

We expect readers to be familiar with the basics of RL, which can be found in Sutton and Barto (2018). RL methods attempt to solve a sequential decision problem, typically formalised as a **Markov decision process (MDP)** , which is a tuple $(S, A, T, I, \mathcal{R}, \gamma)$ where $S$ is a set of states, $A$ is a set of actions, $T : S \times A \to \Delta(S)$ is a transition function, $I \in \Delta(S)$ is an initial state distribution, $\mathcal{R}$ is a reward function, the most general form of which is $\mathcal{R} : S \times A \times S \to \Delta(\mathbb{R})$, and $\gamma \in [0, 1]$ is the discount factor. Here $\Delta(X)$ is the set of all distributions over $X$. A **stationary policy** is a function $\pi : S \to \Delta(A)$ that specifies a distribution over actions in each state, and a **non-stationary** policy is a function $\pi : (S \times A)^* \times S \to \Delta(A)$, where $*$ is the Kleene star. A **trajectory** $\tau$ is a path $s_0, a_0, r_0, ...$ through the MDP that is possible according to $T$, $I$, and $\mathcal{R}$. The **return** of a trajectory is the discounted sum of rewards $G(\tau) \doteq \sum_{t=0}^{\infty} \gamma^t r_t$, and the **value** of a policy is the expected return $J(\pi) \doteq \mathbb{E}_{\tau \sim \pi}[G(\tau)]$. We derive **policy (preference) orderings** from reward functions by ordering policies according to their value. In this paper, we assume that $S$ and $A$ are finite, that $|A| > 1$, that all states are reachable, and that $\mathcal{R}(s, a, s')$ has finite mean for all $s, a, s'$.

In our work, we consider various reward functions for a given environment, which is then formally a **Markov decision process without reward** $MDP \setminus \mathcal{R} \doteq (S, A, T, I, \_\_, \gamma)$. Having fixed an $MDP \setminus \mathcal{R}$, any reward function can be viewed as a function of only the current state and action by marginalizing over transitions: $\mathcal{R}(s, a) \doteq \sum_{s' \sim T(s'|s,a)} \mathcal{R}(s, a, s')$, we adopt this view from here on. We define the **(discounted) visit counts** of a policy as $\mathcal{F}^\pi(s, a) \doteq \mathbb{E}_{\tau \sim \pi}[\sum_{i=0}^{\infty} \gamma^i \mathbb{1}(s_i = s, a_i = a)]$. Note that $J(\pi) = \sum_{s,a} \mathcal{R}(s, a)\mathcal{F}^\pi(s, a)$, which we also write as $\langle \mathcal{R}, \mathcal{F}^\pi \rangle$. When considering multiple reward functions in an $MDP \setminus \mathcal{R}$, we define $J_\mathcal{R}(\pi) \doteq \langle \mathcal{R}, \mathcal{F}^\pi \rangle$ and sometimes use

---

[3]Their result concerns non-stationary policies and use non-Markovian reward functions, but in Appendix C, we show how an analogous construction can be used with stationary policies and Markovian rewards.

$J_i(\pi) \doteq \langle \mathcal{R}_i, \mathcal{F}^\pi \rangle$ as shorthand. We also use $\mathcal{F} : \Pi \to \mathbb{R}^{|S||A|}$ to denote the embedding of policies into Euclidean space via their visit counts, and define $\mathcal{F}(\dot{\Pi}) \doteq \{\mathcal{F}(\pi : \pi \in \dot{\Pi})\}$ for any $\dot{\Pi}$. Moreover, we also use a second way to embed policies into Euclidean space; let $\mathcal{G}(\pi)$ be the $|S||A|$-dimensional vector where $\mathcal{G}(\pi)[s, a] = \pi(a \mid s)$, and let $\mathcal{G}(\dot{\Pi}) \doteq \{\mathcal{G}(\pi : \pi \in \dot{\Pi})\}$.

## 4.2 Definitions and Basic Properties of Hackability and Simplification

Here, we formally define *hackability* as a binary relation between reward functions.

**Definition 1.** A pair of reward functions $\mathcal{R}_1$, $\mathcal{R}_2$ are **hackable** relative to policy set $\Pi$ and an environment $(S, A, T, I, \_\_, \gamma)$ if there exist $\pi, \pi' \in \Pi$ such that

$$J_1(\pi) < J_1(\pi') \ \& \ J_2(\pi) > J_2(\pi'),$$

else they are **unhackable**.

Note that an unhackable reward pair can have $J_1(\pi) < J_1(\pi') \ \& \ J_2(\pi) = J_2(\pi')$ or vice versa. Unhackability is symmetric; this can be seen be swapping $\pi$ and $\pi'$ in Definition 1. It is not transitive, however. In particular, the constant reward function is unhackable with respect to any other reward function, so if it *were* transitive, any pair of policies would be unhackable. Additionally, we say that $\mathcal{R}_1$ and $\mathcal{R}_2$ are **equivalent** on a set of policies $\Pi$ if $J_1$ and $J_2$ induce the same ordering of $\Pi$, and that $\mathcal{R}$ is **trivial** on $\Pi$ if $J(\pi) = J(\pi')$ for all $\pi, \pi' \in \Pi$. It is clear that $\mathcal{R}_1$ and $\mathcal{R}_2$ are unhackable whenever they are equivalent, or one of them is trivial, but this is relatively uninteresting. Our central question is if and when there are other unhackable reward pairs.

The symmetric nature of this definition is counter-intuitive, given that our motivation distinguishes the proxy and true reward functions. We might break this symmetry by only considering policy sequences that monotonically increase the proxy, however, this is equivalent to our original definition of hackability: think of $\mathcal{R}_1$ as the proxy, and consider the sequence $\pi, \pi'$. We could also restrict ourselves to policies that are approximately optimal according to the proxy; Corollary 2 shows that Theorem 1 applies regardless of this restriction. Finally, we define *simplification* as an asymmetric special-case of unhackability; Theorem 3 shows this is in fact a more demanding condition.

**Definition 2.** $\mathcal{R}_2$ is a **simplification** of $\mathcal{R}_1$ relative to policy set $\Pi$ if for all $\pi, \pi' \in \Pi$,

$$J_1(\pi) < J_1(\pi') \implies J_2(\pi) \le J_2(\pi') \ \& \ J_1(\pi) = J_1(\pi') \implies J_2(\pi) = J_2(\pi')$$

and there exist $\pi, \pi' \in \Pi$ such that $J_2(\pi) = J_2(\pi')$ but $J_1(\pi) \ne J_1(\pi')$. Moreover, if $\mathcal{R}_2$ is trivial then we say that this is a **trivial simplification**.

Intuitively, while unhackability allows replacing inequality with equality – or vice versa – a simplification can only replace inequalities with equality, collapsing distinctions between policies. When $\mathcal{R}_1$ is a simplification of $\mathcal{R}_2$, we also say that $\mathcal{R}_2$ is a **refinement** of $\mathcal{R}_1$. We denote this relationship as $\mathcal{R}_1 \unlhd \mathcal{R}_2$ or $\mathcal{R}_2 \unrhd \mathcal{R}_1$ ; the narrowing of the triangle at $R_1$ represents the collapsing of distinctions between policies. If $\mathcal{R}_1 \unlhd \mathcal{R}_2 \unrhd \mathcal{R}_3$, then we have that $\mathcal{R}_1, \mathcal{R}_3$ are unhackable,[4] but if $\mathcal{R}_1 \unrhd \mathcal{R}_2 \unlhd \mathcal{R}_3$, then this is not necessarily the case.[5]

Note that these definitions are given relative to some $MDP \setminus \mathcal{R}$, although we often assume the environment in question is clear from context and suppress this dependence. The dependence on the policy set $\Pi$, on the other hand, plays a critical role in our results.

## 5 Results

Our results are aimed at understanding when it is possible to have an unhackable proxy reward function. We first establish (in Section 5.1) that (non-trivial) unhackability is impossible when considering the set of all policies. We might imagine that restricting ourselves to a set of sufficiently good (according to the proxy) policies would remove this limitation, but we show that this is not the case. We then analyze finite policy sets (with deterministic policies as a special case), and establish necessary and sufficient conditions for unhackability and simplification. Finally, we demonstrate via example that non-trivial simplifications are also possible for some infinite policy sets in Section 5.3.

---

[4] If $J_3(\pi) > J_3(\pi')$ then $J_2(\pi) > J_2(\pi')$, since $\mathcal{R}_2 \unrhd \mathcal{R}_3$, and if $J_2(\pi) > J_2(\pi')$ then $J_1(\pi) \ge J_1(\pi')$, since $\mathcal{R}_1 \unlhd \mathcal{R}_2$. It is therefore not possible that $J_3(\pi) > J_3(\pi')$ but $J_1(\pi) < J_1(\pi')$.

[5] Consider the case where $\mathcal{R}_2$ is trivial – then $\mathcal{R}_1 \unrhd \mathcal{R}_2 \unlhd \mathcal{R}_3$ for any $\mathcal{R}_1, \mathcal{R}_3$.

## 5.1 Non-trivial Unhackability Requires Restricting the Policy Set

We start with a motivating example. Consider the setting shown in Figure 3, where the agent can move left/stay-still/right and gets a reward depending on its state. Let the Gaussian (blue) be the true reward $\mathcal{R}_1$ and the step function (orange) be the proxy $\mathcal{R}_2$. These are hackable. To see this, consider being at state $B$. Let $\pi(B)$ travel to $A$ or $C$ with 50/50 chance, and compare with the policy $\pi'$ that stays at $B$. Then we have that $J_1(\pi) > J_1(\pi')$ and $J_2(\pi) < J_2(\pi')$.

Generally, we might hope that some environments allow for unhackable reward pairs that are not equivalent or trivial. Here we show that this is not the case, unless we impose restrictions on the set of policies we consider.

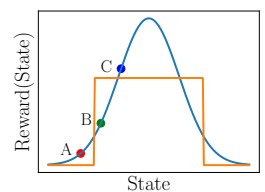

Figure 3: Two reward functions. While the step function may seem like a simplification of the Gaussian, these reward functions are hackable.

First note that if we consider *non-stationary* policies, this result is relatively straightforward. Suppose $\mathcal{R}_1$ and $\mathcal{R}_2$ are *unhackable* and *non-trivial* on the set $\Pi^N$ of all non-stationary policies, and let $\pi^\star$ be a policy that maximises ($\mathcal{R}_1$ and $\mathcal{R}_2$) reward, and $\pi_\perp$ be a policy that *minimises* ($\mathcal{R}_1$ and $\mathcal{R}_2$) reward. Then the policy $\pi_\lambda$ that plays $\pi^\star$ with probability $\lambda$ and $\pi_\perp$ with probability $1 - \lambda$ is a policy in $\Pi^N$. Moreover, for any $\pi$ there are two unique $\alpha, \beta \in [0, 1]$ such that $J_1(\pi) = J_1(\pi_\alpha)$ and $J_2(\pi) = J_2(\pi_\beta)$. Now, if $\alpha \neq \beta$, then either $J_1(\pi) < J_1(\pi_\delta)$ and $J_2(\pi) > J_2(\pi_\delta)$, or vice versa, for $\delta = (\alpha + \beta)/2$. If $\mathcal{R}_1$ and $\mathcal{R}_2$ are unhackable then this cannot happen, so it must be that $\alpha = \beta$. This, in turn, implies that $J_1(\pi) = J_1(\pi')$ iff $J_2(\pi) = J_2(\pi')$, and so $\mathcal{R}_1$ and $\mathcal{R}_2$ are *equivalent*. This means that no interesting unhackability can occur on the set of all non-stationary policies.

The same argument cannot be applied to the set of *stationary* policies, because $\pi_\lambda$ is typically not stationary, and mixing stationary policies' action probabilities does not have the same effect. For instance, consider a hallway environment where an agent can either move left or right. Mixing the "always go left" and "always go right" policies corresponds to picking a direction and sticking with it, whereas mixing their action probabilities corresponds to choosing to go left or right independently at every time-step. However, we will see that there still cannot be any interesting unhackability on this policy set, and, more generally, that there cannot be any interesting unhackability on any set of policies which contains an *open subset*. Formally, a set of (stationary) policies $\dot{\Pi}$ is open if $\mathcal{G}(\dot{\Pi})$ is open in the smallest affine space that contains $\mathcal{G}(\Pi)$, for the set of all stationary policies $\Pi$. We will use the following lemma:

**Lemma 1.** *In any $MDP \setminus \mathcal{R}$, if $\dot{\Pi}$ is an open set of policies, then $\mathcal{F}(\dot{\Pi})$ is open in $\mathbb{R}^{|S|(|A|-1)}$, and $\mathcal{F}$ is a homeomorphism between $\mathcal{G}(\dot{\Pi})$ and $\mathcal{F}(\dot{\Pi})$.*

Using this lemma, we can show that interesting unhackability is impossible on any set of stationary policies $\hat{\Pi}$ which contains an open subset $\dot{\Pi}$. Roughly, if $\mathcal{F}(\dot{\Pi})$ is open, and $\mathcal{R}_1$ and $\mathcal{R}_2$ are non-trivial and unhackable on $\dot{\Pi}$, then the fact that $J_1$ and $J_2$ have a linear structure on $\mathcal{F}(\hat{\Pi})$ implies that $\mathcal{R}_1$ and $\mathcal{R}_2$ must be equivalent on $\dot{\Pi}$. From this, and the fact that $\mathcal{F}(\dot{\Pi})$ is open, it follows that $\mathcal{R}_1$ and $\mathcal{R}_2$ are equivalent everywhere.

**Theorem 1.** *In any $MDP \setminus \mathcal{R}$, if $\hat{\Pi}$ contains an open set, then any pair of reward functions that are unhackable and non-trivial on $\hat{\Pi}$ are equivalent on $\hat{\Pi}$.*

Since simplification is a special case of unhackability, this also implies that non-trivial simplification is impossible for any such policy set. Also note that Theorem 1 makes *no assumptions* about the transition function, etc. From this result, we can show that interesting unhackability always is impossible on the set $\Pi$ of all (stationary) policies. In particular, note that the set $\tilde{\Pi}$ of all policies that always take each action with positive probability is an open set, and that $\tilde{\Pi} \subset \Pi$.

**Corollary 1.** *In any $MDP \setminus \mathcal{R}$, any pair of reward functions that are unhackable and non-trivial on the set of all (stationary) policies $\Pi$ are equivalent on $\Pi$.*

Theorem 1 can also be applied to many other policy sets. For example, we might not care about the hackability resulting from policies with low proxy reward, as we would not expect a sufficiently good learning algorithm to learn such policies. This leads us to consider the following definition:

**Definition 3.** A (stationary) policy $\pi$ is $\varepsilon$-suboptimal if $J(\pi) \geq J(\pi^\star) - \varepsilon$.

Alternatively, if the learning algorithm always uses a policy that is "nearly" deterministic (but with some probability of exploration), then we might not care about hackability resulting from very stochastic policies, leading us to consider the following definition:

**Definition 4.** A (stationary) policy $\pi$ is $\delta$-deterministic if $\forall s \in S \; \exists a \in A : \mathbb{P}(\pi(s) = a) \geq \delta$.

Unfortunately, both of these sets contain open subsets, which means they are subject to Theorem 1.

**Corollary 2.** *In any $MDP \setminus \mathcal{R}$, any pair of reward functions that are unhackable and non-trivial on the set of all $\varepsilon$-suboptimal policies ($\varepsilon > 0$) $\Pi^\varepsilon$ are equivalent on $\Pi^\varepsilon$, and any pair of reward functions that are unhackable and non-trivial on the set of all $\delta$-deterministic policies ($\delta < 1$) $\Pi^\delta$ are equivalent on $\Pi^\delta$.*

Intuitively, Theorem 1 can be applied to any policy set with "volume" in policy space.

## 5.2   Finite Policy Sets

Having established that interesting unhackability is impossible relative to the set of all policies, we now turn our attention to the case of *finite* policy sets. Note that this includes the set of all deterministic policies, since we restrict our analysis to finite MDPs. Surprisingly, here we find that non-trivial non-equivalent unhackable reward pairs *always* exist.

**Theorem 2.** *For any $MDP \setminus \mathcal{R}$, any finite set of policies $\hat{\Pi}$ containing at least two $\pi, \pi'$ such that $\mathcal{F}(\pi) \neq \mathcal{F}(\pi')$, and any reward function $\mathcal{R}_1$, there is a non-trivial reward function $\mathcal{R}_2$ such that $\mathcal{R}_1$ and $\mathcal{R}_2$ are unhackable but not equivalent.*

This proof proceeds by finding a path from $\mathcal{R}_1$ to another reward function $\mathcal{R}_3$ that is hackable with respect to $\mathcal{R}_1$. Along the way to reversing one of $\mathcal{R}_1$'s inequalities, we must encounter a reward function $\mathcal{R}_2$ that instead replaces it with equality. In the case that $\dim(\hat{\Pi}) = 3$, we can visualize moving along this path as rotating the contour lines of a reward function defined on the space containing the policies' discounted state-action occupancies, see Figure 4. This path can be constructed so as to avoid any reward functions that produce trivial policy orderings, thus guaranteeing $\mathcal{R}_2$ is non-trivial. For a *simplification* to exist, we require some further conditions, as established by the following theorem:

**Theorem 3.** *Let $\hat{\Pi}$ be a finite set of policies, and $\mathcal{R}_1$ a reward function. The following procedure determines if there exists a non-trivial simplification of $\mathcal{R}_1$ in a given $MDP \setminus \mathcal{R}$:*

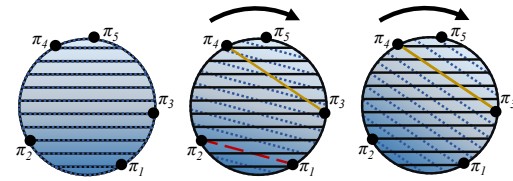

Figure 4: An illustration of the state-action occupancy space with a reward function defined over it. Points correspond to policies' state-action occupancies. Shading intensity indicates expected reward. Rotating the reward function to make $J(\pi_3) > J(\pi_4)$ passes through a reward function that sets $J(\pi_1) = J(\pi_2)$. Solid black lines are contour lines of the original reward function, dotted blue lines are contour lines of the rotated reward function.

1. *Let $E_1 \ldots E_m$ be the partition of $\hat{\Pi}$ where $\pi, \pi'$ belong to the same set iff $J(\pi) = J(\pi')$.*

2. *For each such set $E_i$, select a policy $\pi_i \in E_i$ and let $Z_i$ be the set of vectors that is obtained by subtracting $\mathcal{F}(\pi_i)$ from each element of $\mathcal{F}(E_i)$.*

*Then there is a non-trivial simplification of $\mathcal{R}$ iff $\dim(Z_1 \cup \cdots \cup Z_m) \leq \dim(\mathcal{F}(\hat{\Pi})) - 2$, where $\dim(S)$ is the number of linearly independent vectors in $S$.*

The proof proceeds similarly to Theorem 2. However, in Theorem 2 it was sufficient to show that there are no trivial reward functions along the path from $\mathcal{R}_1$ to $\mathcal{R}_3$, whereas here we additionally need that if $J(\pi) = J(\pi')$ then $J'(\pi) = J'(\pi')$ for all functions $\mathcal{R}_2$ on the path — this is what the extra conditions ensure.

Theorem 3 is opaque, but intuitively, the cases where $\mathcal{R}_1$ cannot be simplified are those where $\mathcal{R}_1$ imposes many different equality constraints that are difficult to satisfy simultaneously. We can think

of $\dim(\mathcal{F}(\Pi))$ as measuring how diverse the behaviours of policies in policy set $\Pi$ are. Having a less diverse policy set means that a given policy ordering imposes fewer constraints on the reward function, creating more potential for simplification. The technical conditions of this proof determine when the diversity of $\Pi$ is or is not sufficient to prohibit simplification, as measured by $\dim(Z_1 \cup \cdots \cup Z_m)$.

Projecting $E_i$ to $Z_i$ simply moves these spaces to the origin, so that we can compare the directions in which they vary (i.e. their span). By assumption, $E_i \cap E_j = \{\}$, but $\mathrm{span}(Z_i) \cap \mathrm{span}(Z_j)$ will include the origin, and may also contain linear subspaces of dimension greater than 0. This is the case exactly when there are a pair of policies in $E_i$ and a pair of policies in $E_j$ that differ by the same visit counts, for example, when the environment contains an obstacle that could be circumnavigated in several different ways (with an impact on visit counts, but no impact on reward), and the policies in $E_i$ and $E_j$ both need to circumnavigate it before doing something else. Roughly speaking, $\dim(Z_1 \cup \cdots \cup Z_m)$ is large when either (i) we have very large and diverse sets of policies in $\hat{\Pi}$ that get the same reward according to $\mathcal{R}$, or (ii) we have a large number of different sets of policies that get the same reward according to $\mathcal{R}$, and where there are different kinds of diversity in the behaviour of the policies in each set. There are also intuitive special cases of Theorem 3. For example, as noted before, if $E_i$ is a singleton then $Z_i$ has no impact on $\dim(Z_1 \cup \cdots \cup Z_m)$. This implies the following corollary:

**Corollary 3.** *For any finite set of policies $\hat{\Pi}$, any environment, and any reward function $\mathcal{R}$, if $|\hat{\Pi}| \geq 2$ and $J(\pi) \neq J(\pi')$ for all $\pi, \pi' \in \hat{\Pi}$ then there is a non-trivial simplification of $\mathcal{R}$.*

A natural question is whether any reward function is guaranteed to have a non-trivial simplification on the set of all deterministic policies. As it turns out, this is not the case. For concreteness, and to build intuition for this result, we examine the set of deterministic policies in a simple $MDP \setminus \mathcal{R}$ with $S = \{0, 1\}, A = \{0, 1\}, T(s, a) = a, I = \{0 : 0.5, 1 : 0.5\}, \gamma = 0.5$. Denote $\pi_{ij}$ the policy that takes action $i$ from state 0 and action $j$ from state 1. There are exactly four deterministic policies. We find that of the $4! = 24$ possible policy orderings, 12 are realizable via some reward function. In each of those 12 orderings, exactly two policies (of the six available pairs of policies in the ordering) can be set to equal value without resulting in the trivial reward function (*which* pair can be equated depends on the ordering in consideration). Attempting to set three policies equal always results in the trivial reward simplification.

For example, given the ordering $\pi_{00} \leq \pi_{01} \leq \pi_{11} \leq \pi_{10}$, the simplification $\pi_{00} = \pi_{01} < \pi_{11} < \pi_{10}$ is represented by $R = \left[\begin{smallmatrix} 0 & 3 \\ 2 & 1 \end{smallmatrix}\right]$, where $\mathcal{R}(s, a) = R[s, a]$: for example, here taking action 1 from state 0 gives reward $\mathcal{R}(0, 1) = 3$. But there is no reward function representing a non-trivial simplification of this ordering with $\pi_{01} = \pi_{11}$. We develop and release a software suite to compute these results. Given an environment and a set of policies, it can calculate all policy orderings represented by some reward function. Also, for a given policy ordering, it can calculate all nontrivial simplifications and reward functions that represent them. For a link to the repository, as well as a full exploration of these policies, orderings, and simplifications, see Appendix D.3.

### 5.3 Unhackability in Infinite Policy Sets

The results in Section 5.1 do not characterize unhackability for infinite policy sets that do not contain open sets. Here, we provide two examples of such policy sets; one of them admits unhackable reward pairs and the other does not. Consider policies $A, B, C$, and reward functions $\mathcal{R}_1$ with $J_1(C) < J_1(B) < J_1(A)$ and $\mathcal{R}_2$ with $J_2(C) = J_2(B) < J_2(A)$. Policy sets $\Pi_a = \{A\} \cup \{\lambda B + (1 - \lambda)C : \lambda \in [0, 1]\}$ and $\Pi_b = \{A\} \cup \{\lambda B + (1 - \lambda)C : \lambda \in [0, 1]\}$ are depicted in Figure 5; the vertical axis represents policies' values according to $\mathcal{R}_1$ and $\mathcal{R}_2$. For $\Pi_a$, $\mathcal{R}_2$ is a simplification of $\mathcal{R}_1$, but for $\Pi_b$, it is not, since $J_1(X) < J_1(Y)$ and $J_2(X) > J_2(Y)$.

## 6 Discussion

We reflect on our results and identify limitations in Section 6.1. In Section 6.2, we discuss how our work can inform discussions about the appropriateness, potential risks, and limitations of using of reward functions as specifications of desired behavior.

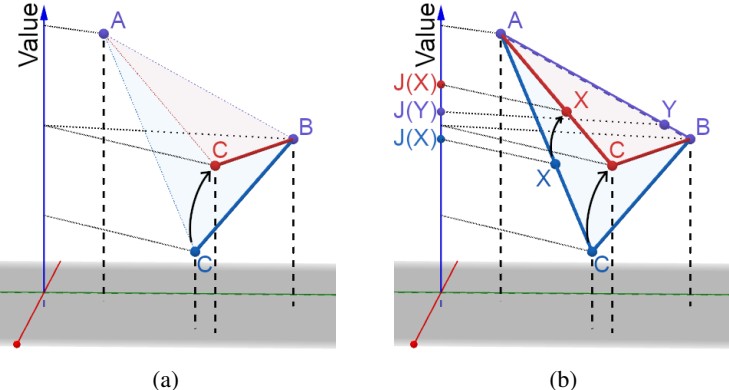

(a)                      (b)

Figure 5: Infinite policy sets that do not contain open sets sometimes allow simplification (a), but not always (b). Points A, B, C represent deterministic policies, while the bold lines between them represent stochastic policies. The y-axis gives the values of the policies according to reward functions $\mathcal{R}_1$ and $\mathcal{R}_2$. We attempt to simplify $\mathcal{R}_1$ by rotating the reward function such that $J_2(B) = J_2(C)$; in the figure, we instead (equivalently) rotate the triangle along the AB axis, leading to the red triangle. In (a), $\mathcal{R}_2$ simplifies $\mathcal{R}_1$, setting all policies along the BC segment equal in value (but still lower than A). In (b), $\mathcal{R}_2$ swaps the relative value of policies X and Y ($J_1(X) < J_1(Y) = J_2(Y) < J_2(X)$) and so does not simplify $\mathcal{R}_1$.

## 6.1   Limitations

Our work has a number of limitations. We have only considered finite MDPs and Markov reward functions, leaving more general environments for future work. While we characterized hackability and simplification for finite policy sets, the conditions for simplification are somewhat opaque, and our characterization of infinite policy sets remains incomplete.

As previously discussed, our definition of hackability is strict, arguably too strict. Nonetheless, we believe that understanding the consequences of this strict definition is an important starting point for further theoretical work in this area.

The main issue with the strictness of our definition has to do with the symmetric nature of hackability. The existence of complex behaviors that yield low proxy reward and high true reward is much less concerning than the reverse, as these behaviors are unlikely to be discovered while optimizing the proxy. For example, it is very unlikely that our agent would solve climate change in the course of learning how to wash dishes. Note that the existence of *simple* behaviors with low proxy reward and high true reward *is* concerning; these could arise early in training, leading us to trust the proxy, only to later see the true reward decrease as the proxy is further optimized. To account for this issue, future work should explore more realistic assumptions about the probability of encountering a given sequence of policies when optimizing the proxy, and measure hackability in proportion to this probability.

We could allow for approximate unhackability by only considering pairs of policies ranked differently by the true and proxy reward functions as evidence of hacking iff their value according to the true reward function differs by more than some $\varepsilon$. Probabilistic unhackability could be defined by looking at the number of misordered policies; this would seem to require making assumptions about the probability of encountering a given policy when optimizing the proxy.

Finally, while unhackability is a guarantee that no hacking will occur, *hackability* is far from a guarantee of hacking. Extensive empirical work is necessary to better understand the factors that influence the occurrence and severity of reward hacking in practice.

## 6.2   Implications

How should we specify our preferences for AI systems' behavior? And how detailed a specification is required to achieve a good outcome? In reinforcement learning, the goal of maximizing (some) reward function is often taken for granted, but a number of authors have expressed reservations about

this approach (Gabriel, 2020; Dobbe et al., 2021; Hadfield-Menell et al., 2016b, 2017; Bostrom, 2014). Our work has several implications for this discussion, although we caution against drawing any strong conclusions due to the limitations mentioned in Section 6.1.

One source of confusion and disagreement is the role of the reward function; it is variously considered as a means of specifying a task (Leike et al., 2018) or encoding broad human values (Dewey, 2011); such distinctions are discussed by Christiano (2019) and Gabriel (2020). We might hope to use Markov reward functions to specify narrow tasks without risking behavior that goes against our broad values. However, if we consider the "narrow task" reward function as a proxy for the true "broad values" reward function, our results indicate that this is not possible: these two reward functions will invariably be hackable. Such reasoning suggests that reward functions must instead encode broad human values, or risk being hacked. This seems challenging, perhaps intractably so, indicating that alternatives to reward optimization may be more promising. Potential alternatives include imitation learning (Ross et al., 2011), constrained RL (Szepesvári, 2020), quantilizers (Taylor, 2016), and incentive management (Everitt et al., 2019).

Scholars have also criticized the assumption that human values can be encoded as rewards (Dobbe et al., 2021), and challenged the use of metrics more broadly (O'Neil, 2016; Thomas and Uminsky, 2022), citing Goodhart's Law (Manheim and Garrabrant, 2018; Goodhart, 1975). A concern more specific to the optimization of reward functions is power-seeking (Turner et al., 2021; Bostrom, 2012; Omohundro, 2008). Turner et al. (2021) prove that optimal policies tend to seek power in most MDPs and for most reward functions. Such behavior could lead to human disempowerment; for instance, an AI system might disable its off-switch (Hadfield-Menell et al., 2016a). Bostrom (2014) and others have argued that power-seeking makes even slight misspecification of rewards potentially catastrophic, although this has yet to be rigorously established.

Despite such concerns, approaches to specification based on learning reward functions remain popular (Fu et al., 2017; Stiennon et al., 2020; Nakano et al., 2021). So far, reward hacking has usually been avoidable in practice, although some care must be taken (Stiennon et al., 2020). Proponents of such approaches have emphasized the importance of learning a reward model in order to exceed human performance and generalize to new settings (Brown et al., 2020a; Leike et al., 2018). But our work indicates that such learned rewards are almost certainly hackable, and so cannot be safely optimized. Thus we recommend viewing such approaches as a means of learning a policy in a safe and controlled setting, which should then be validated before being deployed.

# 7 Conclusion

Our work begins the formal study of reward hacking in reinforcement learning. We formally define hackability and simplification of reward functions, and show conditions for the (non-)existence of non-trivial examples of each. We find that unhackability is quite a strict condition, as the set of all policies never contains non-trivial unhackable pairs of reward functions. Thus in practice, reward hacking must be prevented by limiting the set of possible policies, or controlling (e.g. limiting) optimization. Alternatively, we could pursue approaches not based on optimizing reward functions.

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
