# A  Overview

Section B contains proofs of the main theoretical results. Section D expands on examples given in the main text. Section E presents an unhackability diagram for a generic set of three policies $a, b, c$; Section F shows a simplification diagram of the same policies.

# B  Proofs

Before proving our results, we restate assumptions and definitions. First, recall the preliminaries from Section 4.1, and in particular, that we use $\mathcal{F} : \Pi \to \mathbb{R}^{|S||A|}$ to denote the embedding of policies into Euclidean space via their discounted state-action visit counts, i.e.;

$$\mathcal{F}(\pi)[s, a] = \sum_{t=0}^{\infty} \gamma^t \mathbb{P}(S_t = s, A_t = a).$$

Given a reward function $\mathcal{R}$, let $\vec{\mathcal{R}} \in \mathbb{R}^{|S||A|}$ be the vector where $\vec{\mathcal{R}}[s, a] = \mathbb{E}_{S' \sim T(s,a)}[\mathcal{R}(s, a, S')]$. Note that $J(\pi) = \mathcal{F}(\pi) \cdot \vec{\mathcal{R}}$.

We say $\mathcal{R}_1$ and $\mathcal{R}_2$ are **equivalent** on a set of policies $\Pi$ if $J_1$ and $J_2$ induce the same ordering of $\Pi$, and that $\mathcal{R}$ is **trivial** on $\Pi$ if $J(\pi) = J(\pi')$ for all $\pi, \pi' \in \Pi$. We also have the following definitions from Sections 4 and 5:

**Definition 1.** A pair of reward functions $\mathcal{R}_1$, $\mathcal{R}_2$ are **hackable** relative to policy set $\Pi$ and an environment $(S, A, T, I, \_, \gamma)$ if there exist $\pi, \pi' \in \Pi$ such that

$$J_1(\pi) < J_1(\pi') \ \& \ J_2(\pi) > J_2(\pi'),$$

else they are **unhackable**.

**Definition 2.** $\mathcal{R}_2$ is a **simplification** of $\mathcal{R}_1$ relative to policy set $\Pi$ if for all $\pi, \pi' \in \Pi$,

$$J_1(\pi) < J_1(\pi') \implies J_2(\pi) \leq J_2(\pi') \ \& \ J_1(\pi) = J_1(\pi') \implies J_2(\pi) = J_2(\pi')$$

and there exist $\pi, \pi' \in \Pi$ such that $J_2(\pi) = J_2(\pi')$ but $J_1(\pi) \neq J_1(\pi')$. Moreover, if $\mathcal{R}_2$ is trivial then we say that this is a **trivial simplification**.

Note that these definitions only depend on the policy orderings associated with $\mathcal{R}_2$ and $\mathcal{R}_1$, and so we can (and do) also speak of (ordered) pairs of policy orderings being simplifications or hackable. We also make use of the following definitions:

**Definition 3.** A (stationary) policy $\pi$ is $\varepsilon$-**suboptimal** if $J(\pi) \geq J(\pi^\star) - \varepsilon$, where $\varepsilon > 0$

**Definition 4.** A (stationary) policy $\pi$ is $\delta$-**deterministic** if $\forall s \in S \ \exists a \in A : \mathbb{P}(\pi(s) = a) \geq \delta$, where $\delta < 1$.

## B.1  Non-trivial Unhackability Requires Restricting the Policy Set

Formally, a set of (stationary) policies $\dot{\Pi}$ is **open** if $\mathcal{V}(\dot{\Pi})$ is open in the smallest affine space that contains $\mathcal{V}(\Pi)$, where $\Pi$ is the set of all stationary policies. Note that this space is $|S|(|A| - 1)$-dimensional, since all action probabilities sum to 1.

We require two more propositions for the proof of this lemma.

**Proposition 1.** *If $\dot{\Pi}$ is open then $\mathcal{F}$ is injective on $\dot{\Pi}$.*

*Proof.* First note that, since $\pi(a \mid s) \geq 0$, we have that if $\dot{\Pi}$ is open then $\pi(a \mid s) > 0$ for all $s, a$ for all $\pi \in \dot{\Pi}$. In other words, all policies in $\dot{\Pi}$ take each action with positive probability in each state.

Now suppose $\mathcal{F}(\pi) = \mathcal{F}(\pi')$ for some $\pi, \pi' \in \tilde{\Pi}$. Next, define $w_\pi$ as

$$w_\pi(s) = \sum_{t=0}^{\infty} \gamma^t \mathbb{P}_{\tau \sim \pi}(S_t = s).$$

Note that if $\mathcal{F}(\pi) = \mathcal{F}(\pi')$ then $w_\pi = w_{\pi'}$, and moreover that

$$\mathcal{F}(\pi)[s,a] = w_\pi(s)\pi(a \mid s).$$

Next, since $\pi$ takes each action with positive probability in each state, we have that $\pi$ visits every state with positive probability. This implies that $w_\pi(s) \neq 0$ for all $s$, which means that we can express $\pi$ as

$$\pi(a \mid s) = \frac{\mathcal{F}(\pi)[s,a]}{w_\pi(s)}.$$

This means that if $\mathcal{F}(\pi) = \mathcal{F}(\pi')$ for some $\pi, \pi' \in \tilde{\Pi}$ then $\pi = \pi'$.  $\square$

Note that $\mathcal{F}$ is *not* injective on $\Pi$; if there is some state $s$ that $\pi$ reaches with probability 0, then we can alter the behaviour of $\pi$ at $s$ without changing $\mathcal{F}(\pi)$. But every policy in an open policy set $\dot{\Pi}$ visits every state with positive probability, which then makes $\mathcal{F}$ injective. In fact, Proposition 1 straightforwardly generalises to the set of all policies that visit all states with positive probability (although this will not be important for our purposes).

**Proposition 2.** *Im$(\mathcal{F})$ is located in an affine subspace with $|S|(|A|-1)$ dimensions.*

*Proof.* To show that $\text{Im}(\mathcal{F})$ is located in an affine subspace with $|S|(|A|-1)$ dimensions, first note that

$$\sum_{s,a} \mathcal{F}(\pi)[s,a] = \sum_{t=0}^\infty \gamma^t = \frac{1}{1-\gamma}$$

for all $\pi$. That is, $\text{Im}(\mathcal{F})$ is located in an affine space of points with a fixed $\ell_1$-norm, and this space does not contain the origin.

Next, note that $J(\pi) = \mathcal{F}(\pi) \cdot \vec{\mathcal{R}}$. This means that if knowing the value of $J$ for all $\pi$ determines $\vec{\mathcal{R}}$ modulo at least $n$ free variables, then $\text{Im}(\mathcal{F})$ contains at most $|S||A| - n$ linearly independent vectors. Next recall *potential shaping* (Ng et al., 1999). In brief, given a reward function $\mathcal{R}$ and a *potential function* $\Phi : S \to \mathbb{R}$, we can define a *shaped reward function* $\mathcal{R}'$ by

$$\mathcal{R}'(s,a,s') = \mathcal{R}(s,a,s') + \gamma\Phi(s') - \Phi(s),$$

or, alternatively, if we wish $\mathcal{R}'$ to be defined over the domain $S \times A$,

$$\mathcal{R}'(s,a) = \mathcal{R}(s,a) + \gamma\mathbb{E}_{S'\sim T(s,a)}[\Phi(S')] - \Phi(s).$$

In either case, it is possible to show that if $\mathcal{R}'$ is produced by shaping $\mathcal{R}$ with $\Phi$, and $\mathbb{E}_{S_0 \sim I}[\Phi(S_0)] = 0$, then $J(\pi) = J'(\pi)$ for all $\pi$. This means that knowing the value of $J(\pi)$ for all $\pi$ determines $\vec{\mathcal{R}}$ modulo at least $|S| - 1$ free variables, which means that $\text{Im}(\mathcal{F})$ contains at most $|S||A| - (|S|-1) = |S|(|A|-1) + 1$ linearly independent vectors. Since the smallest affine space that contains $\text{Im}(\mathcal{F})$ does *not* contain the origin, this in turn means that $\text{Im}(\mathcal{F})$ is located in an affine subspace with $= |S|(|A|-1) + 1 - 1 = |S|(|A|-1)$ dimensions.  $\square$

**Lemma 1.** *In any $MDP \setminus \mathcal{R}$, if $\dot{\Pi}$ is an open set of policies, then $\mathcal{F}(\dot{\Pi})$ is open in $\mathbb{R}^{|S|(|A|-1)}$, and $\mathcal{F}$ is a homeomorphism between $\mathcal{V}(\dot{\Pi})$ and $\mathcal{F}(\dot{\Pi})$.*

*Proof.* By the Invariance of Domain Theorem, if

1. $U$ is an open subset of $\mathbb{R}^n$, and

2. $f : U \to \mathbb{R}^n$ is an injective continuous map,

then $f(U)$ is open in $\mathbb{R}^n$ and $f$ is a homeomorphism between $U$ and $f(U)$. We will show that $\mathcal{F}$ and $\dot{\Pi}$ satisfy the requirements of this theorem.

We begin by noting that $\dot{\Pi}$ can be represented as a set of points in $\mathbb{R}^{|S|(|A|-1)}$. First, project $\dot{\Pi}$ into $\mathbb{R}^{|S||A|}$ via $\mathcal{V}$. Next, since $\sum_{a \in A} \pi(a \mid s) = 1$ for all $s$, $\text{Im}(\mathcal{V})$ is in fact located in an affine subspace with $|S|(|A|-1)$ dimensions, which directly gives a representation in $\mathbb{R}^{|S|(|A|-1)}$. Concretely, this represents each policy $\pi$ as a vector $\mathcal{V}(\pi)$ with one entry containing the value $\pi(a \mid s)$ for each

state-action pair $s, a$, but with one action left out for each state, since this value can be determined from the remaining values. We will assume that $\dot{\Pi}$ is embedded in $\mathbb{R}^{|S|(|A|-1)}$ in this way.

By assumption, $\mathcal{V}(\dot{\Pi})$ is an open set in $\mathbb{R}^{|S|(|A|-1)}$. Moreover, by Proposition 2, we have that $\mathcal{F}$ is (isomorphic to) a mapping $\dot{\Pi} \to \mathbb{R}^{|S|(|A|-1)}$. By Proposition 1, we have that $\mathcal{F}$ is injective on $\dot{\Pi}$. Finally, $\mathcal{F}$ is continuous; this can be seen from its definition. We can therefore apply the Invariance of Domain Theorem, and obtain that $\mathcal{F}(\dot{\Pi})$ is open in $\mathbb{R}^{|S|(|A|-1)}$, and that $\mathcal{F}$ is a homeomorphism between $\mathcal{V}(\dot{\Pi})$ and $\mathcal{F}(\dot{\Pi})$. $\qquad\square$

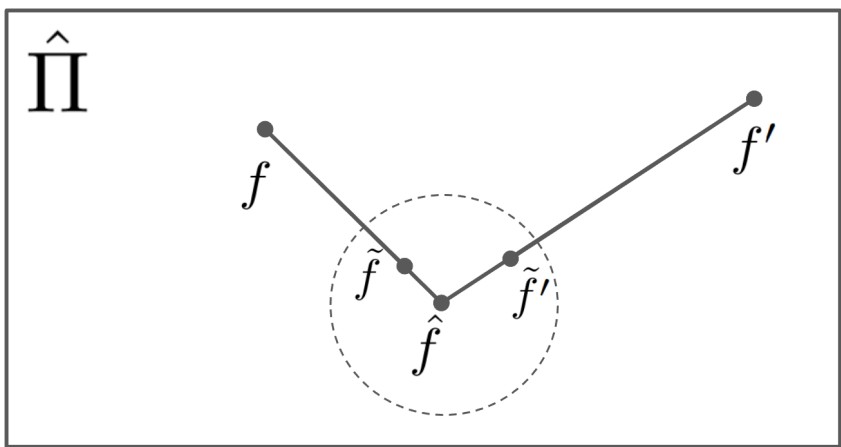

Figure 6: Illustration of the various realizable feature counts used in the proof of Theorem 1.

**Theorem 1.** *In any $MDP \setminus \mathcal{R}$, if $\hat{\Pi}$ contains an open set, then any pair of reward functions that are unhackable and non-trivial on $\hat{\Pi}$ are equivalent on $\hat{\Pi}$.*

*Proof.* Let $\mathcal{R}_1$ and $\mathcal{R}_2$ be any two unhackable and non-trivial reward functions. We will show that, for any $\pi, \pi' \in \hat{\Pi}$, we have $J_1(\pi) = J_1(\pi') \implies J_2(\pi) = J_2(\pi')$, and thus, by symmetry, $J_1(\pi) = J_1(\pi') \iff J_2(\pi) = J_2(\pi')$. Since $\mathcal{R}_1$ and $\mathcal{R}_2$ are unhackable, this further means that they have exactly the same policy order, i.e. that they are equivalent.

Choose two arbitrary $\pi, \pi' \in \hat{\Pi}$ with $J_1(\pi) = J_1(\pi')$ and let $f \doteq \mathcal{F}(\pi), f' \doteq \mathcal{F}(\pi')$. The proof has 3 steps:

1. We find analogues for $f$ and $f'$, $\tilde{f}$ and $\tilde{f}'$, within the same open ball in $\mathcal{F}(\hat{\Pi})$.

2. We show that the tangent hyperplanes of $\vec{R}_1$ and $\vec{R}_2$ at $\tilde{f}$ must be equal to prevent neighbors of $\tilde{f}$ from making $\mathcal{R}_1$ and $\mathcal{R}_2$ hackable.

3. We use linearity to show that this implies that $J_2(\pi) = J_2(\pi')$.

**Step 1:** By assumption, $\hat{\Pi}$ contains an open set $\dot{\Pi}$. Let $\hat{\pi}$ be some policy in $\dot{\Pi}$, and let $\hat{f} \doteq \mathcal{F}(\hat{\pi})$. Since $\dot{\Pi}$ is open, Lemma 1 implies that $\mathcal{F}(\dot{\Pi})$ is open in $\mathbb{R}^{|S|(|A|-1)}$. This means that, if $v, v'$ are the vectors such that $\hat{f} + v = f$ and $\hat{f} + v' = f'$, then there is a positive but sufficiently small $\delta$ such that $\tilde{f} \doteq \hat{f} + \delta v$ and $\tilde{f}' \doteq \hat{f} + \delta v'$ both are located in $\mathcal{F}(\dot{\Pi})$, see Figure 6. This further implies that there are policies $\tilde{\pi}, \tilde{\pi}' \in \dot{\Pi}$ such that $\mathcal{F}(\tilde{\pi}) = \tilde{f}$ and $\mathcal{F}(\tilde{\pi}') = \tilde{f}'$.

**Step 2:** Recall that $J(\pi) = \mathcal{F}(\pi) \cdot \vec{\mathcal{R}}$. Since $\mathcal{R}_1$ is non-trivial on $\hat{\Pi}$, it induces a $(|S|(|A|-1)-1)$-dimensional hyperplane tangent to $\vec{\mathcal{R}}_1$ corresponding to all points $x \in \mathbb{R}^{|S|(|A|-1)}$ such that $x \cdot \vec{\mathcal{R}}_1 = \tilde{f} \cdot \vec{\mathcal{R}}_1$, and similarly for $\mathcal{R}_2$. Call these hyperplanes $H_1$ and $H_2$, respectively. Note that $\tilde{f}$ is contained in both $H_1$ and $H_2$.

Next suppose $H_1 \neq H_2$. Then, we would be able to find a point $f_{12} \in \mathcal{F}(\dot{\Pi})$, such that $f_{12} \cdot \vec{\mathcal{R}}_1 > \tilde{f} \cdot \vec{\mathcal{R}}_1$ but $f_{12} \cdot \vec{\mathcal{R}}_2 < \tilde{f} \cdot \vec{\mathcal{R}}_2$. This, in turn, means that there is a policy $\pi_{12} \in \dot{\Pi}$ such that $\mathcal{F}(\pi_{12}) = f_{12}$,

and such that $J_1(\pi_{12}) > J_1(\tilde{\pi})$ but $J_2(\pi_{12}) < J_2(\tilde{\pi})$. Since $\mathcal{R}_1$ and $\mathcal{R}_2$ are unhackable, this is a contradiction. Thus $H_1 = H_2$.

**Step 3:** Since $J_1(\pi) = J_1(\pi')$, we have that $f \cdot \vec{\mathcal{R}}_1 = f' \cdot \vec{\mathcal{R}}_1$. By linearity, this implies that $\tilde{f} \cdot \vec{\mathcal{R}}_1 = \tilde{f}' \cdot \vec{\mathcal{R}}_1$; we can see this by expanding $\tilde{f} = \hat{f} + \delta v$ and $\tilde{f}' = \hat{f} + \delta v'$. This means that $\tilde{f}' \in H_1$. Now, since $H_1 = H_2$, this means that $\tilde{f}' \in H_2$, which in turn implies that $\tilde{f} \cdot \vec{\mathcal{R}}_2 = \tilde{f}' \cdot \vec{\mathcal{R}}_2$. By linearity, this then further implies that $f \cdot \vec{\mathcal{R}}_2 = f' \cdot \vec{\mathcal{R}}_2$, and hence that $J_2(\pi) = J_2(\pi')$. Since $\pi, \pi'$ were chosen arbitrarily, this means that $J_1(\pi) = J_1(\pi') \implies J_2(\pi) = J_2(\pi')$. $\qquad \square$

**Corollary 1.** *In any $MDP \setminus \mathcal{R}$, any pair of reward functions that are unhackable and non-trivial on the set of all (stationary) policies $\Pi$ are equivalent on $\Pi$.*

*Proof.* This corollary follows from Theorem 1, if we note that the set of all policies does contain an open set. This includes, for example, the set of all policies in an $\epsilon$-ball around the policy that takes all actions with equal probability in each state. $\qquad \square$

**Corollary 2.** *In any $MDP \setminus \mathcal{R}$, any pair of reward functions that are unhackable and non-trivial on the set of all $\varepsilon$-suboptimal policies ($\varepsilon > 0$) $\Pi^\varepsilon$ are equivalent on $\Pi^\varepsilon$, and any pair of reward functions that are unhackable and non-trivial on the set of all $\delta$-deterministic policies ($\delta < 1$) $\Pi^\delta$ are equivalent on $\Pi^\delta$.*

*Proof.* To prove this, we will establish that both $\Pi^\varepsilon$ and $\Pi^\delta$ contain open policy sets, and then apply Theorem 1.

Let us begin with $\Pi^\delta$. First, let $\pi$ be some deterministic policy, and let $\pi_\epsilon$ be the policy that in each state with probability $1 - \epsilon$ takes the same action as $\pi$, and otherwise samples an action uniformly. Then if $\delta < \epsilon < 1$, $\pi_\epsilon$ is the center of an open ball in $\Pi^\delta$. Thus $\Pi^\delta$ contains an open set, and we can apply Theorem 1.

For $\Pi^\varepsilon$, let $\pi^\star$ be an optimal policy, and apply an analogous argument. $\qquad \square$

## B.2 Finite Policy Sets

**Theorem 2.** *For any $MDP \setminus \mathcal{R}$, any finite set of policies $\hat{\Pi}$ containing at least two $\pi, \pi'$ such that $\mathcal{F}(\pi) \neq \mathcal{F}(\pi')$, and any reward function $\mathcal{R}_1$, there is a non-trivial reward function $\mathcal{R}_2$ such that $\mathcal{R}_1$ and $\mathcal{R}_2$ are unhackable but not equivalent.*

*Proof.* If $\mathcal{R}_1$ is trivial, then simply choose any non-trivial $\mathcal{R}_2$. Otherwise, the proof proceeds by finding a path from $\vec{\mathcal{R}}_1$ to $-\vec{\mathcal{R}}_1$, and showing that there must be an $\vec{\mathcal{R}}_2$ on this path such that $\mathcal{R}_2$ is non-trivial and unhackable with respect to $\mathcal{R}_1$, but not equivalent to $\mathcal{R}_1$.

The key technical difficulty is to show that there exists a continuous path from $\mathcal{R}_1$ to $-\mathcal{R}_1$ in $\mathbb{R}^{|S||A|}$ that does not include any trivial reward functions. Once we've established that, we can simply look for the first place where an inequality is reversed – because of continuity, it first becomes an equality. We call the reward function at that point $\mathcal{R}_2$, and note that $\mathcal{R}_2$ is unhackable wrt $\mathcal{R}_1$ and not equivalent to $\mathcal{R}_1$. We now walk through the technical details of these steps.

First, note that $J(\pi) = \mathcal{F}(\pi) \cdot \vec{\mathcal{R}}$ is continuous in $\vec{\mathcal{R}}$. This means that if $J_1(\pi) > J_2(\pi')$ then there is a unique first vector $\vec{\mathcal{R}}_2$ on any path from $\vec{\mathcal{R}}_1$ to $-\vec{\mathcal{R}}_1$ such that $\mathcal{F}(\pi) \cdot \vec{\mathcal{R}}_2 \not> \mathcal{F}(\pi) \cdot \vec{\mathcal{R}}_2$, and for this vector we have that $\mathcal{F}(\pi) \cdot \vec{\mathcal{R}}_2 = \mathcal{F}(\pi) \cdot \vec{\mathcal{R}}_2$. Since $\hat{\Pi}$ is finite, and since $\mathcal{R}_1$ is not trivial, this means that on any path from $\vec{\mathcal{R}}_1$ to $-\vec{\mathcal{R}}_1$ there is a unique first vector $\vec{\mathcal{R}}_2$ such that $\mathcal{R}_2$ is not equivalent to $\mathcal{R}_1$, and then $\mathcal{R}_2$ must also be a unhackable with respect to $\mathcal{R}_1$.

It remains to show that there is a path from $\vec{\mathcal{R}}_1$ to $-\vec{\mathcal{R}}_1$ such that no vector along this path corresponds to a trivial reward function. Once we have such a path, the argument above implies that $\mathcal{R}_2$ must be a non-trivial reward function that is unhackable with respect to $\mathcal{R}_1$. We do this using a dimensionality argument. If $\mathcal{R}$ is trivial on $\hat{\Pi}$, then there is some $c \in \mathbb{R}$ such that $\mathcal{F}(\pi) \cdot \vec{\mathcal{R}} = c$ for all $\pi \in \hat{\Pi}$. This means that if $\mathcal{F}(\hat{\Pi})$ has at least $d$ linearly independent vectors, then the set of all such vectors $\vec{\mathcal{R}}$ forms a linear subspace with at most $|S||A| - d$ dimensions. Now, since $\hat{\Pi}$ contains at least two $\pi, \pi'$ such that $\mathcal{F}(\pi) \neq \mathcal{F}(\pi')$, we have that $\mathcal{F}(\hat{\Pi})$ has at least 2 linearly independent vectors, and

hence that the set of all reward functions that are trivial on $\hat{\Pi}$ forms a linear subspace with at most $|S||A| - 2$ dimensions. This means that there must exist a path from $\vec{\mathcal{R}}_1$ to $-\vec{\mathcal{R}}_1$ that avoids this subspace, since only a hyperplane (with dimension $|S||A| - 1$) can split $\mathbb{R}^{|S||A|}$ into two disconnected components. $\qquad\square$

**Theorem 3.** *Let $\hat{\Pi}$ be a finite set of policies, and $\mathcal{R}$ a reward function. The following procedure determines if there exists a non-trivial simplification of $\mathcal{R}$ in a given $MDP \setminus \mathcal{R}$:*

1. *Let $E_1 \ldots E_m$ be the partition of $\hat{\Pi}$ where $\pi, \pi'$ belong to the same set iff $J(\pi) = J(\pi')$.*

2. *For each such set $E_i$, select a policy $\pi_i \in E_i$ and let $Z_i$ be the set of vectors that is obtained by subtracting $\mathcal{F}(\pi_i)$ from each element of $\mathcal{F}(E_i)$.*

*Then there is a non-trivial simplification of $\mathcal{R}$ iff $\dim(Z_1 \cup \cdots \cup Z_m) \leq \dim(\mathcal{F}(\hat{\Pi})) - 2$, where $\dim(S)$ is the number of linearly independent vectors in $S$.*

*Proof.* This proof uses a similar proof strategy as Theorem 2. However, in addition to avoiding trivial reward functions on the path from $\vec{\mathcal{R}}_1$ to $-\vec{\mathcal{R}}_1$, we must also ensure that we stay within the "equality-preserving space", to be defined below.

First recall that $\mathcal{F}(\hat{\Pi})$ is a set of vectors in $\mathbb{R}^{|S||A|}$. If $\dim(\mathcal{F}(\hat{\Pi})) = D$ then these vectors are located in a $D$-dimensional linear subspace. Therefore, we will consider $\mathcal{F}(\hat{\Pi})$ to be a set of vectors in $\mathbb{R}^D$. Next, recall that any reward function $\mathcal{R}$ induces a linear function $L$ on $\mathbb{R}^D$, such that $J = L \circ \mathcal{F}$, and note that there is a $D$-dimensional vector $\vec{\mathcal{R}}$ that determines the *ordering* that $\mathcal{R}$ induces over all points in $\mathbb{R}^D$. To determine the *values* of $J$ on all points in $\mathbb{R}^D$ we would need a $(D+1)$-dimensional vector, but to determine the *ordering*, we can ignore the height of the function. In other words, $L(x) = x \cdot \vec{\mathcal{R}} + L(\vec{0})$, for any $x \in \mathbb{R}^D$. Note that this is a different vector representation of reward functions than that which was used in Theorem 2 and before.

Suppose $\mathcal{R}_2$ is a reward function such that if $J_1(\pi) = J_1(\pi')$ then $J_2(\pi) = J_2(\pi')$, for all $\pi, \pi' \in \hat{\Pi}$. This is equivalent to saying that $L_2(\mathcal{F}(\pi)) = L_2(\mathcal{F}(\pi'))$ if $\pi, \pi' \in E_i$ for some $E_i$. By the properties of linear functions, this implies that if $\mathcal{F}(E_i)$ contains $d_i$ linearly independent vectors then it specifies a $(d_i - 1)$-dimensional affine space $S_i$ such that $L_2(x) = L_2(x')$ for all points $x, x' \in S_i$. Note that this is the smallest affine space which contains all points in $E_i$. Moreover, $L_2$ is also constant for any affine space $\bar{S}_i$ *parallel* to $S_i$. Formally, we say that $\bar{S}_i$ is parallel to $S_i$ if there is a vector $z$ such that for any $y \in \bar{S}_i$ there is an $x \in S_i$ such that $y = x + z$. From the properties of linear functions, if $L_2(x) = L_2(x')$ then $L_2(x + z) = L_2(x' + z)$.

Next, from the transitivity of equality, if we have two affine spaces $\bar{S}_i$ and $\bar{S}_j$, such that $L_2$ is constant over each of $\bar{S}_i$ and $\bar{S}_j$, and such that $\bar{S}_i$ and $\bar{S}_j$ *intersect*, then $L_2$ is constant over all points in $\bar{S}_i \cup \bar{S}_j$. From the properties of linear functions, this then implies that $L_2$ is constant over all points in the smallest affine space $\bar{S}_i \otimes \bar{S}_j$ containing $\bar{S}_i$ and $\bar{S}_j$, given by combining the linearly independent vectors in $\bar{S}_i$ and $\bar{S}_j$. Note that $\bar{S}_i \otimes \bar{S}_j$ has between $\max(d_i, d_j)$ and $(d_i + d_j - 1)$ dimensions. In particular, since the affine spaces of $Z_1 \ldots Z_m$ intersect (at the origin), and since $L_2$ is constant over these spaces, we have that $L_2$ must be constant for all points in the affine space $\mathcal{Z}$ which is the smallest affine space containing $Z_1 \cup \cdots \cup Z_m$. That is, if $\mathcal{R}_2$ is a reward function such that $J_1(\pi) = J_1(\pi') \implies J_2(\pi) = J_2(\pi')$ for all $\pi, \pi' \in \hat{\Pi}$, then $L_2$ is constant over $\mathcal{Z}$. Moreover, if $L_2$ is constant over $\mathcal{Z}$ then $L_2$ is also constant over each of $E_1 \ldots E_m$, since each of $E_1 \ldots E_m$ is parallel to $\mathcal{Z}$. This means that $\mathcal{R}_2$ satisfies that $J_1(\pi) = J_1(\pi') \implies J_2(\pi) = J_2(\pi')$ for all $\pi, \pi' \in \hat{\Pi}$ if and only if $L_2$ is constant over $\mathcal{Z}$.

If $\dim(\mathcal{Z}) = D'$ then there is a linear subspace with $D - D'$ dimensions, which contains the ($D$-dimensional) vector $\vec{\mathcal{R}}_2$ of any reward function $\mathcal{R}_2$ where $J_1(\pi) = J_1(\pi') \implies J_2(\pi) = J_2(\pi')$ for $\pi, \pi' \in \hat{\Pi}$. This is because $\mathcal{R}_2$ is constant over $\mathcal{Z}$ if and only if $\vec{R}_2 \cdot v = 0$ for all $v \in \mathcal{Z}$. Then if $\mathcal{Z}$ contains $D'$ linearly independent vectors $v_i \ldots v_{D'}$, then the solutions to the corresponding system of linear equations form a $(D - D')$ dimensional subspace of $\mathbb{R}^D$. Call this space the *equality-preserving space*. Next, note that $\mathcal{R}_2$ is trivial on $\hat{\Pi}$ if and only if $\vec{\mathcal{R}}_2$ is the zero vector $\vec{0}$.

Now we show that if the conditions are not satisfied, then there is no non-trivial simplification. Suppose $D' \geq D - 1$, and that $\mathcal{R}_2$ is a simplification of $\mathcal{R}_1$. Note that if $\mathcal{R}_2$ simplifies $\mathcal{R}_1$ then

$\vec{\mathcal{R}}_2$ is in the equality-preserving space. Now, if $D' = D$ then $L_2$ (and $L_1$) must be constant for all points in $\mathbb{R}^D$, which implies that $\mathcal{R}_2$ (and $\mathcal{R}_1$) are trivial on $\hat{\Pi}$. Next, if $D' = D - 1$ then the equality-preserving space is one-dimensional. Note that we can always preserve all equalities of $\mathcal{R}_1$ by *scaling* $\mathcal{R}_1$ by a constant factor. That is, if $\mathcal{R}_2 = c \cdot \mathcal{R}_1$ for some (possibly negative) $c \in \mathbb{R}$ then $J_1(\pi) = J_1(\pi') \implies J_2(\pi) = J_2(\pi')$ for all $\pi, \pi' \in \hat{\Pi}$. This means that the parameter which corresponds to the dimension of the equality-preserving space in this case must be the scaling of $\vec{\mathcal{R}}_2$. However, the only simplification of $\mathcal{R}_1$ that is obtainable by uniform scaling is the trivial simplification. This means that if $D' \geq D - 1$ then $\mathcal{R}_1$ has no non-trivial simplifications on $\hat{\Pi}$.

For the other direction, suppose $D' \leq D - 2$. Note that this implies that $\mathcal{R}_1$ is not trivial. Let $\mathcal{R}_3 = -\mathcal{R}_1$. Now both $\vec{\mathcal{R}}_1$ and $\vec{\mathcal{R}}_3$ are located in the equality-preserving space. Next, since the equality-preserving space has at least two dimensions, this means that there is a continuous path from $\vec{\mathcal{R}}_1$ to $\vec{\mathcal{R}}_3$ through the equality-preserving space that does not pass the origin. Now, note that $J_i(\pi) = \mathcal{F}(\pi) \cdot \vec{\mathcal{R}}_i$ is continuous in $\vec{\mathcal{R}}_i$. This means that there, on the path from $\vec{\mathcal{R}}_1$ to $\vec{\mathcal{R}}_3$ is a first vector $\vec{\mathcal{R}}_2$ such that $\mathcal{F}(\pi) \cdot \vec{\mathcal{R}}_2 = \mathcal{F}(\pi') \cdot \vec{\mathcal{R}}_2$ but $\mathcal{F}(\pi) \cdot \vec{\mathcal{R}}_1 \neq \mathcal{F}(\pi') \cdot \vec{\mathcal{R}}_1$ for some $\pi, \pi' \in \hat{\Pi}$. Let $\mathcal{R}_2$ be a reward function corresponding to $\vec{\mathcal{R}}_2$. Since $\vec{\mathcal{R}}_2$ is not $\vec{0}$, we have that $\mathcal{R}_2$ is not trivial on $\hat{\Pi}$. Moreover, since $\vec{\mathcal{R}}_2$ is in the equality-preserving space, and since $\mathcal{F}(\pi) \cdot \vec{\mathcal{R}}_2 = \mathcal{F}(\pi') \cdot \vec{\mathcal{R}}_2$ but $\mathcal{F}(\pi) \cdot \vec{\mathcal{R}}_1 \neq \mathcal{F}(\pi') \cdot \vec{\mathcal{R}}_1$ for some $\pi, \pi' \in \hat{\Pi}$, we have that $\mathcal{R}_2$ is a non-trivial simplification of $\mathcal{R}_1$. Therefore, if $D' \leq D - 2$ then there exists a non-trivial simplification of $\mathcal{R}_1$.

We have thus proven both directions, which completes the proof. $\square$

**Corollary 3.** *For any finite set of policies $\hat{\Pi}$, any environment, and any reward function $\mathcal{R}$, if $|\hat{\Pi}| \geq 2$ and $J(\pi) \neq J(\pi')$ for all $\pi, \pi' \in \hat{\Pi}$, then there is a non-trivial simplification of $\mathcal{R}$.*

*Proof.* Note that if $E_i$ is a singleton set then $Z_i = \{\vec{0}\}$. Hence, if each $E_i$ is a singleton set then $\dim(Z_1 \cup \cdots \cup Z_m) = 0$. If $\hat{\Pi}$ contains at least two $\pi, \pi'$, and $J(\pi) \neq J(\pi')$, then $\mathcal{F}(\pi) \neq \mathcal{F}(\pi')$. This means that $\dim(\mathcal{F}(\hat{\Pi})) \geq 2$. Thus the conditions of Theorem 3 are satisfied. $\square$

## C  Any Policy Can Be Made Optimal

In this section, we show that any policy is optimal under some reward function.

**Proposition 3.** *For any rewardless MDP $(S, A, T, I, \_\_, \gamma)$ and any policy $\pi$, there exists a reward function $\mathcal{R}$ such that $\pi$ is optimal in the corresponding MDP $(S, A, T, I, \mathcal{R}, \gamma)$.*

*Proof.* Let $\mathcal{R}(s, a, s') = 0$ if $a \in \text{Support}(\pi(s))$, and $-1$ otherwise. $\square$

This shows that any policy is rationalised by some reward function in any environment. Any policy that gives 0 probability to any action which $\pi$ takes with 0 probability is optimal under this construction. This means that if $\pi$ is deterministic, then it will be the only optimal policy in $(S, A, T, I, \mathcal{R}, \gamma)$.

## D  Examples

In this section, we take a closer look at two previously-seen examples: the two-state $MDP \setminus \mathcal{R}$ and the cleaning robot.

### D.1  Two-state $MDP \setminus \mathcal{R}$ example

Let us explore in more detail the two-state system introduced in the main text. We decsribe this infinite-horizon $MDP \setminus \mathcal{R}$ in Table 1.

We denote $\pi_{ij}$ $(i, j \in \{0, 1\})$ the policy which takes action $i$ when in state 0 and action $j$ when in state 1. This gives us four possible deterministic policies:

$$\{\pi_{00}, \pi_{01}, \pi_{10}, \pi_{11}\}.$$

| States | $S = \{0, 1\}$ |
|---|---|
| Actions | $A = \{0, 1\}$ |
| Dynamics | $T(s, a) = a$ for $s \in S, a \in A$ |
| Initial state distribution | $\Pr(\text{start in } s) = 0.5$ for $s \in S$ |
| Discount factor | $\gamma = 0.5$ |

Table 1: The two-state $MDP \setminus \mathcal{R}$ in consideration.

There are $4! = 24$ ways of ordering these policies with strict inequalities. Arbitrarily setting $\pi_{00} < \pi_{11}$ breaks a symmetry and reduces the number of policy orderings to 12. When a policy ordering can be derived from some reward function $\mathcal{R}$, we say that $\mathcal{R}$ **represents** it, and that the policy ordering is **representable**. Of these 12 policy orderings with strict inequalities, six are representable:

$$\pi_{00} < \pi_{01} < \pi_{10} < \pi_{11},$$
$$\pi_{00} < \pi_{01} < \pi_{11} < \pi_{10},$$
$$\pi_{00} < \pi_{10} < \pi_{01} < \pi_{11},$$
$$\pi_{01} < \pi_{00} < \pi_{11} < \pi_{10},$$
$$\pi_{10} < \pi_{00} < \pi_{01} < \pi_{11},$$
$$\pi_{10} < \pi_{00} < \pi_{11} < \pi_{01}.$$

Simplification in this environment is nontrivial – given a policy ordering, it is not obvious which strict inequalities can be set to equalities such that there is a reward function which represents the new ordering. Through a computational approach (see Section D.3) we find the following representable orderings, each of which is a simplification of one of the above strict orderings.

$$\pi_{00} = \pi_{01} < \pi_{11} < \pi_{10},$$
$$\pi_{00} = \pi_{10} < \pi_{01} < \pi_{11},$$
$$\pi_{00} < \pi_{01} = \pi_{10} < \pi_{11},$$
$$\pi_{01} < \pi_{00} = \pi_{11} < \pi_{10},$$
$$\pi_{10} < \pi_{00} = \pi_{11} < \pi_{01},$$
$$\pi_{00} < \pi_{01} < \pi_{10} = \pi_{11},$$
$$\pi_{10} < \pi_{00} < \pi_{01} = \pi_{11},$$
$$\pi_{00} = \pi_{01} = \pi_{10} = \pi_{11}.$$

Furthermore, for this environment, we find that any reward function which sets the value of three policies equal necessarily forces the value of the fourth policy to be equal as well.

## D.2 Cleaning robot example

Recall the cleaning robot example in which a robot can choose to clean a combination of three rooms, and receives a nonnegative reward for each room cleaned. This setting can be thought of as a single-step eight-armed bandit with special reward structure.

### D.2.1 Hackability

We begin our exploration of this environment with a statement regarding exactly when two policies are hackable. In fact, the proposition is slightly more general, extending to an arbitrary (finite) number of rooms.

**Proposition 4.** *Consider a cleaning robot which can clean $N$ different rooms, and identify each room with a unique index in $\{1, \ldots, N\}$. Cleaning room $i$ gives reward $r(i) \geq 0$. Cleaning multiple rooms gives reward equal to the sum of the rewards of the rooms cleaned. The value of a policy $\pi_S$ which cleans a collection of rooms $S$ is the sum of the rewards corresponding to the rooms cleaned: $J(\pi_S) = \sum_{i \in S} r(i)$. For room $i$, the true reward function assigns a value $r_{true}(i)$, while the proxy reward function assigns it reward $r_{proxy}(i)$. The proxy reward is hackable with respect to the true*

*reward if and only if there are two sets of rooms $S_1, S_2$ such that $\sum_{i \in S_1} r_{proxy}(i) < \sum_{i \in S_2} r_{proxy}(i)$ and $\sum_{i \in S_1} r_{true}(i) > \sum_{i \in S_2} r_{true}(i)$.*

*Proof.* We show the two directions of the double implication.

$\Leftarrow$ Suppose there are two sets of rooms $S_1, S_2$ satisfying $\sum_{i \in S_1} r_{proxy}(i) < \sum_{i \in S_2} r_{proxy}(i)$ and $\sum_{i \in S_1} r_{true}(i) > \sum_{i \in S_2} r_{true}(i)$. The policies $\pi_{S_i} =$ "clean exactly the rooms in $S_i$" for $i \in \{1, 2\}$ demonstrate that $r_{proxy}, r_{true}$ are hackable. To see this, remember that $J(\pi_S) = \sum_{i \in S} r(i)$. Combining this with the premise immediately gives $J_{proxy}(\pi_{S_1}) < J_{proxy}(\pi_{S_2})$ and $J_{true}(\pi_{S_1}) > J_{true}(\pi_{S_2})$.

$\Rightarrow$ If $r_{proxy}, r_{true}$ are hackable, then there must be a pair of policies $\pi_1, \pi_2$ such that $J_{proxy}(\pi_1) < J_{proxy}(\pi_2)$ and $J_{true}(\pi_1) > J_{true}(\pi_2)$. Let $S_1$ be the set of rooms cleaned by $\pi_1$ and $S_2$ be the set of rooms cleaned by $\pi_2$. Again remembering that $J(\pi_S) = \sum_{i \in S} r(i)$ immediately gives us that $\sum_{i \in S_1} r_{proxy}(i) < \sum_{i \in S_2} r_{proxy}(i)$ and $\sum_{i \in S_1} r_{true}(i) > \sum_{i \in S_2} r_{true}(i)$.

$\square$

In the main text, we saw two intuitive ways of modifying the reward function in the cleaning robot example: omitting information and overlooking fine details. Unfortunately, there is no obvious mapping of Proposition 4 onto simple rules concerning how to safely omit information or overlook fine details: it seems that one must resort to ensuring that no two sets of rooms satisfy the conditions for hackability described in the proposition.

### D.2.2 Simplification

We now consider simplification in this environment. Since we know the reward for cleaning each room is nonnegative, there will be some structure underneath all the possible orderings over the policies. This structure is shown in Figure 7: regardless of the value assigned to each room, a policy at the tail of an arrow can only be at most as good as a policy at the head of the arrow.

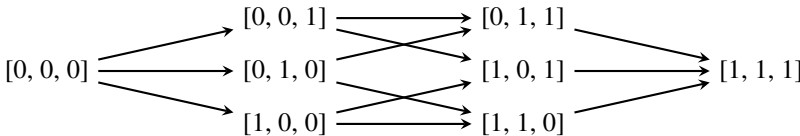

Figure 7: The structure underlying all possible policy orderings (assuming nonnegative room value). The policy at the tail of the arrow is at most as good as the policy at the head of the arrow.

If we decide to simplify an ordering by equating two policies connected by an arrow, the structure of the reward calculation will force other policies to also be equated. Specifically, if the equated policies differ only in position $i$, then all pairs of policies which differ only in position $i$ will also be set equal.

For example, imagine we simplify the reward by saying we don't care if the attic is cleaned or not, so long as the other two rooms are cleaned (recall that we named the rooms Attic, Bedroom and Kitchen). This amounts to saying that $J([0, 1, 1]) = J([1, 1, 1])$. Because the policy value function is of the form

$$J(\pi) = J([x, y, z]) = [x, y, z] \cdot [r_1, r_2, r_3]$$

where $x, y, z \in \{0, 1\}$, this simplification forces $r_1 = 0$. In turn, this implies that $J([0, 0, 0]) = J([1, 0, 0])$ and $J([0, 1, 0]) = J([1, 1, 0])$. The new structure underlying the ordering over policies is shown in Figure 8.

[X, 0, 0] → [X, 0, 1] → [X, 1, 1]
[X, 0, 0] → [X, 1, 0] → [X, 1, 1]

Figure 8: The updated ordering structure after equating "clean all the rooms" and "clean all the rooms except the attic". X can take either value in $\{0, 1\}$.

An alternative way to think about simplification in this problem is by imagining policies as corners of a cube, and simplification as flattening of the cube along one dimension – simplification collapses this cube into a square.

## D.3  Software repository

The software suite described in the paper (and used to calculate the representable policy orderings and simplifications of the two-state $MDP \setminus \mathcal{R}$) can be found at `https://anonymous.4open.science/r/simplified-reward-5130`.

# E  Unhackability Diagram

Consider a setting with three policies $a, b, c$. We allow all possible orderings of the policies. In general, these orderings might not all be representable; a concrete case in which they are is when $a, b, c$ represent different deterministic policies in a 3-armed bandit.

We can represent all unhackable pairs of policy orderings with an undirected graph, which we call an **unhackability diagram**. This includes a node for every representable ordering and edges connecting orderings which are unhackable. Figure 9 shows an unhackability diagram including all possible orderings of the three policies $a, b, c$.

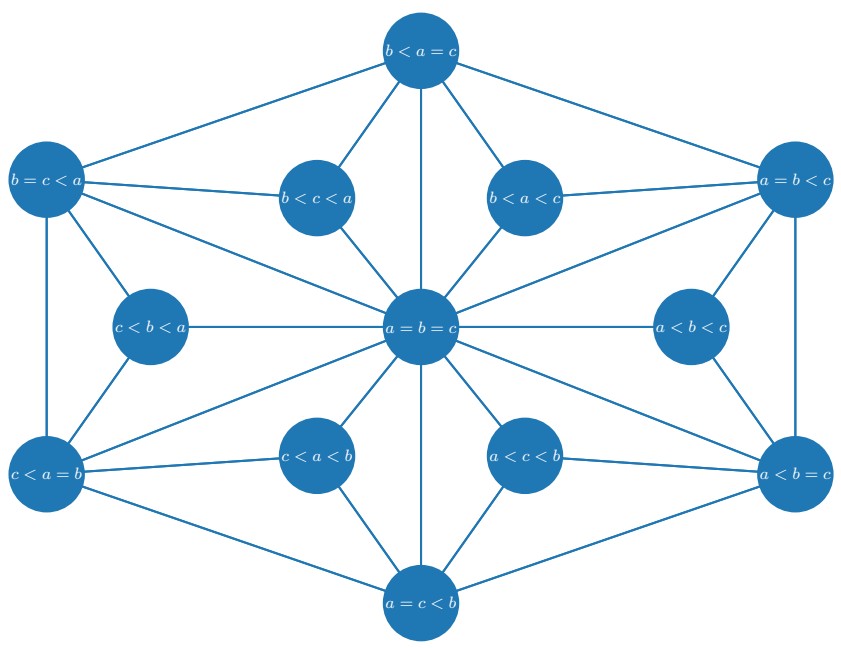

Figure 9: Illustration of the unhackable pairs of policy orderings when considering all possible orderings over three policies $a, b, c$. Edges of the graph connect unhackable policy orderings.

# F   Simplification Diagram

We can also represent all possible simplifications using a directed graph, which we call a **simplification diagram**. This includes a node for every representable ordering and edges pointing from orderings to their simplifications. Figure 10 presents a simplification diagram including all possible orderings of three policies $a, b, c$.

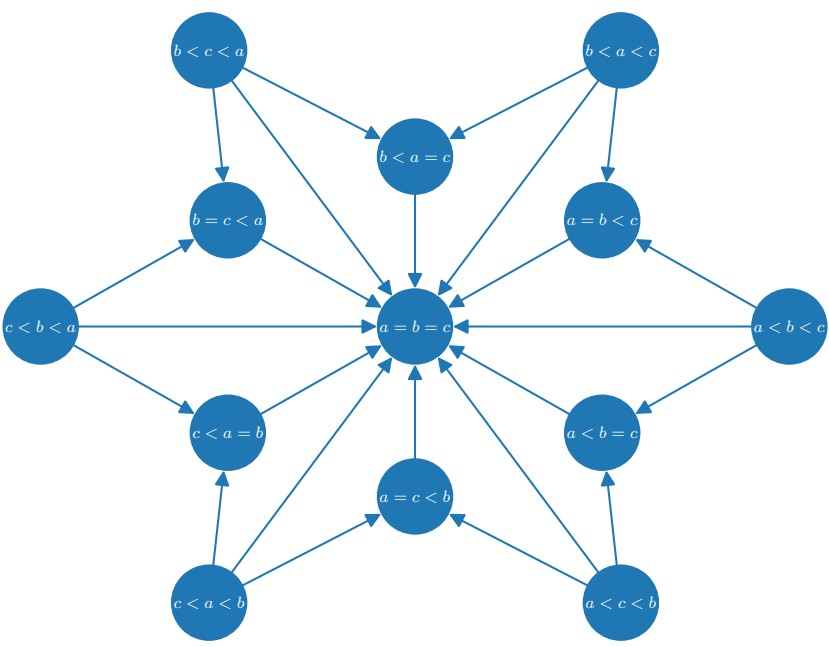

Figure 10: Illustration of the simplifications present when considering all possible orderings over three policies $a, b, c$. Arrows represent simplification: the policy ordering at the head of an arrow is a simplification of the policy ordering at the tail of the arrow.

We note that the simplification graph is a subgraph of the unhackability graph. This will always be the case, since simplification can never lead to reward hacking.