# OpenReview forum: "Defining and Characterizing Reward Gaming"
_NeurIPS.cc/2022/Conference — NeurIPS 2022 Accept_

### Official Review · Reviewer_a6pX · 2022-07-07

**Rating:** 5
**Confidence:** 3
**Soundness:** 3 good
**Presentation:** 2 fair
**Contribution:** 2 fair

**Summary:**

This paper studies the gameability of reward functions in MDPs. The authors define a ungameable reward function as a reward function that induces (almost) the same preferences over policies. In addition, they consider a special case of ungameable reward functions termed simplification. Focus of the theoretical work is the existence of ungameable rewards under the class of stochastic policies as well as finite policy classes, e.g. deterministic policies .

**Questions:**

**Definition of Ungameable Reward Functions:**
If I understand Definition~1 correctly, how you want to define ungameable reward functions is that *two reward function are ungameable if and only if they induce the same preference structure over all policies*. In particular, I think that you should clarify that, if I am not mistaken, your definition of ungameable reward functions is equivalent to: $R_1$ and $R_2$ are ungameable if $\forall \pi, \pi'$: $J_1(\pi) < J_1(\pi') \Rightarrow J_2(\pi) \leq J_2(\pi')$. Again, this definition is extremely restrictive if you look (as you do) at general policy classes. Moreover, this would mean that the difference between ungameable reward functions and simplifications as you define them is very minor. The only difference is the additional requirement of $J_1(\pi) = J_1(\pi') \Rightarrow J_2(\pi) = J_2(\pi')$; if I am not mistaken?

**Definition of Reward Functions:**  In Section~4.1, you define a reward function as a function $R: S \times A \times S \to \Delta(\mathbb{R})$. However, later on you treat $R(s, a, s')$ (and $R(s, a)$ defined by marginalisation) as a scalar, e.g. lines 147, 149. It seems that you're only interested in the expected reward anyways? This is a very convoluted way to express that you're looking at reward functions $R:S\times A \to \mathbb{R}$; if I am not mistaken.



I would be interested in whether this work has any potential practical impact? The procedure in Theorem 3, tests whether there is a non-trivial simplification for a reward function, but why would that be interesting? Also, does the procedure in Theorem 3 only test for the existence, or can it also actually compute a simplification? (I am aware of the section on "implications", but I am curious about something concrete that can be done based on your results. This should also not be taken as a point of critique.)



**Limitations:**

I very much appreciate the authors acknowledging the limitations of their work in their discussion.

**Strengths And Weaknesses:**

The problem considered is important and interesting. Moreover, the obtained theoretical results are interesting, although limited by the considered notion of gameability.

I understand that the authors not only want to look at optimal policies, as it is briefly mention in the introduction. However, looking at \emph{every} possible policy (in some policy class) naturally leads to weak results and is, in my opinion (and as you yourselves note), overly conservative.
The definition of ungameable reward functions essentially requires that the preference structure over policies is (almost) the same ($\pi \prec_{R_1} \pi'$ has to imply $\pi \preceq_{R_2} \pi'$; see my point below).


While I acknowledge that such a restrictive notion of gameable can be a starting point, I must say that, in my opinion, the insights and conclusions that one would potentially take away from the paper could be misleading. You write in your discussion that "But our work indicates that such learned rewards are almost certainly gameable, and so cannot be safely optimized." (line 380), but this is most likely only the case for your very restrictive definition of gameability. To me a more natural starting point would have been to look at specific optimisation frameworks such as policy gradient.

In particular, to me the given example (Figure~1) actually illustrates really well why your definition of ungameable rewards is somewhat irritating. If an agent were to maximise for the proxy [1, 1, 0], then an optimal policy would be [1, 1, 0]. However, the return maximising policy with respect to the true rewards, [1, 1, 1], is actually [1, 1, 1]. Maximising the proxy in fact induces undesired behaviour (not cleaning the 3rd room).

---

> ### Author Response · Authors · 2022-08-02
> **Author Response**
>
> Thank you for your review.
>
> * **Regarding the first weakness (paragraph 2 of the "strengths and weaknesses" section), and the question about the definition of ungameable reward functions:**  First, you are correct that the definition is restrictive.  But it is not immediately obvious just how restrictive it is (as we aim to illustrate via examples); one of our contributions is to rigorously establish precisely how restrictive this notion of gameability is.  We also consider this definition well motivated by the practical concern of avoiding reward gaming.  Note that it is equivalent to the following, which more clearly reflects our motivation: $\mathcal{R_1}$ is gameable wrt $\mathcal{R_2}$ if there exists a sequence of policies
> $\pi_1 <_1 \pi_2 … <_1 \pi_T$ such that $\pi_t >_2 \pi_t'$ for some $t < t'$ (where $<_1, <_2$ are the orderings over policies according to their value under $\mathcal{R_1}, \mathcal{R_2}$).  In other words, if the proxy $\mathcal{R_1}$ is gameable wrt $\mathcal{R_2}$, then (partially) optimizing $\mathcal{R_1}$ is liable to decrease reward wrt $\mathcal{R_2}$ (at some point over training).  The implication is: you might verify that an agent’s behavior is acceptable at time-step t, but if you continue optimizing the proxy, that could lead the agent to perform worse (e.g. unacceptably poorly) with respect to the true reward.  We use this line or reasoning to argue that it may be necessary to validate the safety of a policy trained on a proxy reward at the end of training (line 380-383).  We will revise the paper to include this definition/equivalence and discussion if the review committee thinks this would be valuable.
> * **Regarding the 2nd weakness and the proposal to look at specific optimization frameworks such as policy gradient:** Our results are intended to be general enough to apply to *any* optimization framework, including policy gradient; this is why we make weak assumptions about the sequence of policies found via optimization.  We believe in-depth analysis of specific methods of optimization would be complementary to our work, and is better motivated now that we’ve established the difficulty of preventing gaming solely by finding a good-enough proxy.
> * **Regarding the example in Figure 1:** for $r_{proxy}=[1,1,0]$, the policy [1,1,1] is optimal as well, in addition to [1,1,0].  Thus it is not correct to say “maximizing the proxy induces undesired behavior”, only “maximizing the proxy *could* induce undesirable behavior”; i.e. the proxy does not distinguish between the optimal and the suboptimal behaviors.   We believe it is exactly this distinction that makes our definition of gameability -- and our result about its restrictiveness -- interesting.  Our intuition is that gaming occurs when optimizing the proxy reward function *selects* for changes that reduce the true reward. In this case there is no selection pressure to go from [1,1,1] to [1,1,0] as this won’t increase the proxy reward, so $r_{proxy}=[1,1,0]$ is ungameable w.r.t. $r_{true}=[1,1,1]$.
> * **Regarding the definition of reward functions:** You are correct that we only care about the expected returns (i.e. value) of policues. We include this extra complexity because our results are general enough to cover the cases of reward that is stochastic and/or depends on the next state.  We will remove/change this if the review committee thinks it is distracting or confusing.
> * **Regarding practical impact:** We primarily view our work as a starting point for further development of theory in this area.  Nevertheless, we believe Theorem 1 supports the practical conclusion that proxy reward functions (even those learned from human behavior/preferences) should not be viewed as objectives to be optimized, but only as a means of providing useful training signals to a policy (lines 380-383).  This calls into question the idea that reward modelling “separates learning what to achieve (the ‘What?’) from learning how to achieve it (the ‘How?’)” (Leike et al. 2018, page 2).  Our work demonstrates that such a separation is not possible unless we learn the true reward function (or something equivalent).  An additional practical recommendation from Theorem 2 is to explore restricting the policy set (e.g. to $\epsilon$-greedy policies) as an approach to mitigating reward gaming; we are not aware of this being previously proposed.  Finally, we also provide software to compute ungameability and simplification graphs given a set of deterministic policies as a practical aid for future research.  For instance, this could be used to evaluate various ways of restricting the policy set.

---

> > ### Author Response · Authors · 2022-08-02
> > **[continuing] Author Response**
> >
> > [continuing from above, due to character limit]
> >
> > * **“Does the procedure in Thm 3 only test for existence, or could it also be used to compute a simplification?”:** In principle you could compute a simplification using the method described in the proof of Theorem 3 (ie, modify the reward along a path in the corresponding vector space, which stays within the “equality-preserving space” and at the same time avoids the “trivial space”), although this would be very computationally expensive. However, the point of Theorem 3 is not primarily to provide a method for computing simplifications, but rather to demonstrate that simplification sometimes is possible, and sometimes not, as well as to provide some insight into what kinds of environments allow for simplification.

---

> > ### Author Response · Authors · 2022-08-02
> > **One further comment regarding optimization frameworks and our definition**
> >
> > While above we describe optimization as producing a sequence of policies that monotonically improve according to the proxy, our definition and analysis apply in exactly the same way so long as the final policy produced by optimization has higher value according to the proxy than the initial policy or any intermediary policy of interest.

---

> > ### Comment · Reviewer_a6pX · 2022-08-05
> > **Response to rebuttal**
> >
> > Thank you for your additional explanations.
> >
> > After reading them, I am slightly more convinced of the definition of gameable rewards. In particular, what you wrote in the first paragraph about the motivation of considering a sequence of policies makes a fairly good argument in favour of your definition (even though not all policies would've to be considered for something like this). I personally think that the paper would benefit from adding this early on in the paper.

---

> > > ### Author Response · Authors · 2022-08-05
> > > **We agree and will update the paper accordingly.**
> > >
> > > Thank you for the response.
> > > We consider the definition of gameability one of our main contributions, and agree that we should include further justification for this particular definition being a good one, or even "the right one".
> > > This is something the reader should not have to think through themselves, so we will update the paper to include all of the reasons we've given above for believing this definition is apt.

---

### Official Review · Reviewer_8Thp · 2022-07-08

**Rating:** 8
**Confidence:** 4
**Soundness:** 4 excellent
**Presentation:** 4 excellent
**Contribution:** 4 excellent

**Summary:**

This paper provides a formal characterization of the phenomenon of reward gaming, when optimizing a proxy reward results in poor outcomes according to the true reward. The authors present a formal definition for gameability as a relation between the true reward and a proxy reward: if there is a way to improve a policy according to one reward function that makes it worse according to the other. They also define when a proxy is a simplification of the true reward, as a special case of ungameability. The paper provides theoretical results showing that non-trivial ungameability is impossible when the set of policies has an open subset, and thus requires restricting the policy set. They find that in the case of finite policy sets, non-trivial ungameable reward pairs always exist and non-trivial simplifications exist under certain conditions. They also investigate infinite policy sets and find that it is sometimes possible to find ungameable pairs.


**Questions:**

The implications section discusses that Markov rewards may not be suitable for specifying narrow tasks. This question is explored in depth in "On the Expressivity of Markov Reward" (Abel et al, 2021), which could be a good reference to connect to this discussion.

Do you expect that inverse reward design (Hadfield-Menell et al, 2017) or decoupled approval (Uesato et al, 2020) could also be viable alternatives to reward optimization for the purposes of avoiding reward gaming?

**Limitations:**

The authors have extensively discussed the limitations of this work in the Limitations section.


**Strengths And Weaknesses:**

To my knowledge, this is the first paper to formally characterize the phenomenon of reward gaming for general reward functions. Reward gaming is a widespread phenomenon in AI with over 60 documented examples, which has received some empirical investigation by Pan et al (2021), but has been poorly understood despite its ubiquity. This paper makes significant progress on formally defining reward gaming and answering the question "when is it safe to optimize a proxy reward?". It's a very clear and well-written paper and an important contribution to the field.

The main weakness of this work is that the definition of gameability seems too strict (as the authors acknowledge in the Limitations section), and as a result the existence of ungameable pairs has only been established for finite policy sets, which is a significant constraint. I expect that finding reasonable ways to relax this definition will be a productive area for future work.

---

> ### Author Response · Authors · 2022-08-02
> **Author Response**
>
> Thank you for your review.  Responding to your questions:
>
> * Thanks for noting that we neglected to discuss Abel et al. (2021); we agree this work is highly relevant, and we discuss it in the revision.
> * Inverse reward design learns a distribution over the true reward function’s parameters given a proxy reward; afterwards the authors use a risk-averse planning algorithm to optimize the minimal reward over samples from this distribution.  This risk-averse planning could address gameability if sampled reward functions do not unanimously prefer a worse (according to the true reward) policy.  It would be interesting to explore this further in future work.  It is also an interesting question if there are proxy/true reward pairs for which IRD on the proxy gives you the true reward (when taking the MLE or average of the distribution over reward functions, instead of doing risk-averse planning).
> * Decoupled approval is motivated as an approach to reward *tampering*, which is a distinct problem from reward gaming, and thus seems less likely to be relevant.

---

### Official Review · Reviewer_Tpmi · 2022-07-09

**Rating:** 6
**Confidence:** 2
**Soundness:** 3 good
**Presentation:** 4 excellent
**Contribution:** 3 good

**Summary:**

By giving the formal definition and looking into several cases, this paper initiates the study of reward gaming. Reward gaming is a phenomenon where optimizing with respect to a proxy target will cause some failure on the true target. This paper studies a more general concept of gameable/ungameable in reinforcement learning: for a fixed policy set, a pair of reward functions are gameable if and only if there are two policies which show different value preferences under them. This paper first shows that for non-stationary policies, probabilistic stationary policies, and any policy set which contains an open set, there is only equivalent ungameability. Next, it proves that finite sets are always non-trivially gameable. Finally, the paper shows an example of infinite sets (without open set) to show that ungameability is not definitive.

**Questions:**

What is the definition for $\mathcal{F} (\Pi)$ in Lemma 1 and Theorem 3?
I believe it is $\\{ \mathcal{F} (\pi) \ :\ \pi \in \Pi \\}$, but it seems to be missing in the main text.

**Limitations:**

I think the main limitation is that the concept of gameability is too generic. If the optimization is with respect to $\mathcal{R}_2$, we may believe that $J_2 (\pi) \ge J_2 (\pi^\star)$. This asymmetry is generally satisfied in reality.

**Strengths And Weaknesses:**

Strengths:
1. It is the first paper formally studying the concept of reward gaming. This concept is rather interesting in my view.
2. The notions are basically introduced clearly.
3. Theorem 1 and Theorem 2 are crucial to this paper.


Weaknesses:
1. The property of asymmetry in proxy optimization is not well exploited.
2. I think for Definition 2 and the rotating contour lines proof for Theorem 2, the authors should provide more intuitive explanations.

---

> ### Author Response · Authors · 2022-08-02
> **Author response**
>
> Thank you for your review.
>
> * We provide more intuition for Definition 2 (simplification) in the revision.  Briefly, simplification breaks the symmetry of ungameability: $\mathcal{R_2}$ is an ungameable proxy of $\mathcal{R_1}$, if we can transform $\mathcal{R_1}$'s policy ordering to $\mathcal{R_2}$'s by replacing inequalities in $\mathcal{R_1}$ with equalities in $\mathcal{R_2}$, and/or vice versa.  Simplification restricts this to the first operation.  We call this simplification because the preference ordering of the proxy expresses less preferences (i.e. more indifference) between outcomes, in this sense, it is “simpler”.
> * We also provide more intuition for Theorem 2 in our revision.  The core idea of Theorem 2, illustrated by the rotating plates in Figure 4, is that as we move along a path between (e.g.) $\mathcal{R}$ and $-\mathcal{R}$, we *must* change the ordering of policies at some point, which requires setting two policies equal first (thus yielding an ungameable reward function at this point).  The key technical difficulty here is showing that there is such a path that avoids the trivial reward function (i.e. flattening the plate).  Restricting the path to rotations would be one way of avoiding flattening the plate.
> * We already consider 3 forms of asymmetry in this work: 1) simplification, mentioned above; 2) restricting the policy set to policies that are approximately optimal according to the proxy (Corollary 2 applies whether “approximately optimal” means “according to the true reward” or “according to the proxy”); 3) restricting ourselves to sequences of policies where the proxy increases monotonically (this is suggested by line 35, but we discuss it explicitly in the revision).  Of these, only simplification made a difference in our analysis, indicating that accounting for asymmetry in any meaningful way is non-trivial.  We view this as an important direction for future work (line 330).
> * RE definition of $\mathcal{F}(\Pi)$: you are correct; this is made explicit in the revision.
> * Sorry, we didn’t understand this comment: “If the optimization is with respect to $\mathcal{R_2}$, we may believe that $J_2(\pi) \geq J_2(\pi^*)$”.  Can you please clarify your point to help us respond effectively?

---

> > ### Comment · Reviewer_Tpmi · 2022-08-05
> > **post rebuttal response**
> >
> > Suppose the real reward is $\mathcal{R}_1$ and the value is $J_1$; the proxy reward is $\mathcal{R}_2$ and the value is $J_2$. The optimal policy $\pi^\star$ of the real model satisfies $J_1 (\pi^\star) \ge J_1 (\pi)$ for any $\pi$. However, the optimal solution $\pi'$ found under $\mathcal{R}_2$ satisfies $J_2 (\pi') \ge J_2 (\pi^\star)$. Taking this asymmetry into consideration, can you provide (or do you believe there exists) any results?

---

> > > ### Author Response · Authors · 2022-08-08
> > > **All of our results cover this case, and none are specific to this case.**
> > >
> > > Thank you for elaborating.  It's possible we've still not understood your point, but will respond as best we can anyways...
> > >
> > > Because our definitions of (simplification/)gameability considers not only the optimal policy, but the entire policy order, we don't believe this property is likely to lead to any additional results or novel proof strategies regarding gameabillity.  Since we've already characterized gameability for finite policy sets and sets that contain open balls, this property might prove useful only for characterizing gameability for infinite policy sets not containing open balls.
> > >
> > > There might be some alternative notion similar in spirit to gameability where this property would be relevant.  For instance, we could also consider only policies that are strictly optimal according to the proxy or true reward, but this is beyond the scope of our work.
> > >
> > > Did you have any particular result or type of result in mind, or that you would expect?

---

### Official Review · Reviewer_zA2n · 2022-07-11

**Rating:** 6
**Confidence:** 3
**Soundness:** 3 good
**Presentation:** 3 good
**Contribution:** 2 fair

**Summary:**

This paper formalizes and studies the problem of gameability in RL, where performing better under a proxy results in worse performance under the true reward. The paper provides the first theoretical study of gameability and shows that un-gamable reward functions that are non-trivial is only possible with finite sets of policies.

**Questions:**

What about changes in policy that keep proxy the same but reduce the true reward? Is that a problem? Policy search isn't always monotonic for deep RL so you may still end up reducing the true reward over time.

Lines 80-84: This rule doesn't cover the second unbalanced example above since there is no room that the proxy says is better than the true. Same for simplification. Is this just a sufficient condition?

Line 138: is this a kleeny star? good to define since often star is used for optimal in MDP literature

Figure 3: I don't understand why this motivating example is being given. How does this help the reader in the following sections? How is it related to the theory?

301 "there" is no reward

Figure 5: Very confusing. what are the different colors. what are the axis for J_1 and J_2? What is C'?

Line 358: I don't get this point about narrow tasks. Isn't the same true about specifying human values? Won't simplification often leads to gaming in this scenario as well?

Line 376: "functions remain"

Line 390: If you don't optimize a reward how do you even define gameability?

**Limitations:**

The authors mention limitations several times throughout the paper.

**Strengths And Weaknesses:**

+First rigorous approach to defining gameability

+Interesting and novel theory

+Good motivating example in Fig 1

+I like the idea of looking at optimization and not relying on agents finding optimal policy for proxy

-It is unclear how to make use of the results in this paper. In particular, after reading the paper I do not think that I have any tools for addressing actual gameability problems in RL and it is unclear how much the theoretical results matter in practice.

-More discussion relating equivalent rewards and prior work (e.g. Ng and Russell 2000 and Brown et al. 2020b) on sets of rewards that make a certain policy optimal.

-Some of the writing and results are unclear/opaque.

---

> ### Author Response · Authors · 2022-08-02
> **Author Response**
>
> Thank you for your review.
>
> Responding to Questions:
> * “Changes in policy that keep proxy the same but reduce the true reward” could of course also be problematic, and one could give a stronger definition of ungameability which forbids such cases as well. There are a few reasons for why we have used this weaker definition instead. First, if a proxy reward is gameable in our sense, then a policy optimisation procedure can be actively incentivised to decrease the true reward. We expect this to be more problematic than the case when the proxy reward merely fails to disincentivise reduction of the true reward. Second, using a weaker definition of ungameability makes Theorem 1 stronger, which of course is desirable. Third, note that while policy optimisation can be non-monotonic, it is reasonable to assume that the final policy, after optimisation is performed, will have a higher (proxy) reward than the original policy.  Therefore, our analysis does not require that the entire optimisation path is monotonic --- we simply wish that the policy that is produced by optimising the proxy is not worse than the original policy according to the true reward.
> * The example in Figure 3 motivates section 5.1 which follows right after it, where we show that non-trivial ungameability requires restricting the policy set. Our hope is to give the reader an intuition for how a situation abstractly described in 5.1 might play out in practice.  In the revision, we move this example into section 5.1.
> * Line 80-84: No it is necessary and sufficient; the condition is that there are 2 *sets* of rooms with $r_{true}(S_1) > r_{true}(S_2)$ and $r_{proxy}(S_1) < r_{proxy}(S_2)$.   In this case, those sets are $S_1=\{Kitchen\}$ and $S_2=\{Attic, Bedroom\}$, and we have that $r_{true}(S_1) = 3 > 2.5 = r_{true}(S_2)$, but also that $r_{proxy}(S_1) = 1 < 2 = r_{proxy}(S_2)$, and thus we have that the reward functions are gameable.
> * Line 138: Yes this is a Kleene star; we make this explicit in the revision.
> * Line 358: this was confusingly stated, and we’ve clarified and unpacked it a bit in the revision.  Our point here is as follows: One might imagine that we can use reward functions to specify narrow tasks without risking behavior that goes against our broad values.  If we consider the “narrow task” reward function as a proxy for the true “broad values” reward function, however, our results indicate that this is not possible: these two reward functions will invariably be gameable.
> * Line 390: The idea is that we might avoid the problem of gameability entirely by using a different paradigm than reinforcement learning.
>
> Regarding the weaknesses:
> * We agree that we do not provide tools for addressing reward gaming in practice.  We primarily view our work as a starting point for further theory in this area.  Nevertheless, we believe Theorem 1 supports the practical conclusion that proxy reward functions (even those learned from human behavior/preferences) should not be viewed as objectives to be optimized, but only as a means of providing useful training signals to a policy (lines 380-383).  This calls into question the most ambitious aims of reward modelling.  For instance, Leike et al. (2018; page 2) describe reward modelling as separating “learning what to achieve (the ‘What?’) from learning how to achieve it (the ‘How?’)”.  But our work indicates that such a separation may not be possible in practice.
> * We include a discussion of previous work on equivalence of reward functions in the revision.
> * We fixed the typos (thanks)!
> * We've updated Figure 5 and the caption; please let us know if you think further clarification is needed.

---

> > ### Comment · Reviewer_zA2n · 2022-08-08
> > **response**
> >
> > Thank you for your clarifications and responses.

---

### Public Comment · ~Jakob_Thumm1 · 2023-08-02
**Paper title changed**

It is not clear whether the title of this work should be "Defining and Characterizing Reward Hacking" or "Defining and Characterizing Reward Gaming".
The bibtex and openreview page claim the latter, but the PDF defines the term "reward hacking" throughout the paper (except Figure 1).

---

### Meta-Review · Area_Chair_rGD6 · 2022-08-24

**Recommendation:** Accept
**Confidence:** Certain

**Metareview:**

The paper introduces the question of whether a pair of reward function and a proxy reward function is "gameable", i.e., whether maximizing the proxy may actually decrease the return in the original reward function. The authors develop a fairly complete analysis of this problem revealing a set of non-trivial results on, e.g., the possibility of creating ungameable pairs.

There is a general consensus among the reviewers that the contribution is novel and interesting and thus I'm proposing acceptance. For the final version of the paper, I strongly suggest the authors to
- Integrate some of the discussion from the rebuttal
- Clarify that the scope of the paper is more on the introduction of the concept and it's analysis rather in proposing new algorithms or practical approaches
- Expand as much as possible its connections with existing literature on inverse reinforcement learning

**Award:**

No

---

### Decision · Program_Chairs · 2022-09-14

Accept